# Rational design of a sensitivity-enhanced tracer for discovering efficient APC–Asef inhibitors

Jie Zhong[1,2,3,9], Yuegui Guo[4,9], Shaoyong Lu ®[2,9], Kun Song[2], Ying Wang[2], Li Feng[3], Zhen Zheng[3], Qiufen Zhang[3], Jiacheng Wei[2], Peng Sang[5], Yan Shi[5], Jianfeng Cai ®[5], Guoqiang Chen ®[6], Chen-Ying Liu ®[4] ✉, Xiuyan Yang ®[2,7] ✉ & Jian Zhang ®[1,2,3,8] ✉

The adenomatous polyposis coli (APC)–Rho guanine nucleotide exchange factor 4 (Asef) protein–protein interaction (PPI) is essential for colorectal cancer metastasis, making it a promising drug target. Herein, we obtain a sensitivity-enhanced tracer (tracer 7) with a high binding affinity ($K_d$ = 0.078 μM) and wide signal dynamic range (span = 251 mp). By using tracer 7 in fluorescence-polarization assays for APC–Asef inhibitor screening, we discover a best-in-class inhibitor, MAI-516, with an $IC_{50}$ of 0.041 ± 0.004 μM and a conjugated transcriptional transactivating sequence for generating cell-permeable MAIT-516. MAIT-516 inhibits CRC cell migration by specifically hindering the APC–Asef PPI. Furthermore, MAIT-516 exhibits no cytotoxic effects on normal intestinal epithelial cell and colorectal cancer cell growth. Overall, we develop a sensitivity-enhanced tracer for fluorescence polarization assays, which is used for the precise quantification of high-activity APC–Asef inhibitors, thereby providing insight into PPI drug development.

Protein–protein interactions (PPIs) play a fundamental role in virtually all biological processes and are considered the "holy grail" of modern life science and medicine[1–4]. PPI manipulation has immense potential for drug development, and thus PPIs are emerging as important targets for the pharmaceutical treatment of human diseases. Due to the relatively large, flat, and conformationally featureless characteristics of PPI interfaces, PPI targeting by small-molecule inhibitors is challenging in drug discovery, and PPIs are frequently deemed "undruggable". As an alternative, the development of motif-based peptide or peptidomimetic inhibitors, which mimic critical secondary structures, i.e., the

"hotspots" in PPI interfaces, provides an efficient strategy for drugging PPIs[5–7].

The adenomatous polyposis coli (APC)–Rho guanine nucleotide exchange factor 4 (Asef) PPI represents one of the quintessential examples of the rational design of therapeutic peptide or peptidomimetic inhibitors for clinical implications. APC is responsible for multiple signaling pathways via various protein interactions. Commonly, APC promotes the degradation of β-catenin and thereby negatively regulates Wnt signaling. In colorectal cancer (CRC) metastasis, *APC* gene alterations were observed to generate C-terminal truncated APC

[1]State Key Laboratory of Medical Genomics, National Research Center for Translational Medicine at Shanghai, Ruijin Hospital, Shanghai Jiao Tong University School of Medicine, Shanghai, China. [2]Department of Pathophysiology, Key Laboratory of Cell Differentiation and Apoptosis of Chinese Ministry of Education, Shanghai Jiao Tong University School of Medicine, Shanghai, China. [3]Medicinal Chemistry and Bioinformatics Center, Shanghai Jiao Tong University School of Medicine, Shanghai, China. [4]Department of Colorectal and Anal Surgery, Xinhua Hospital, Shanghai Jiao Tong University School of Medicine, Shanghai, China. [5]Department of Chemistry, University of South Florida, Tampa, FL, USA. [6]Research Unit of Stress and Cancer, Chinese Academy of Medical Sciences, Shanghai, China. [7]State Key Laboratory of Quality Research in Chinese Medicine, Institute of Chinese Medical Sciences, University of Macau, Macau, China. [8]School of Pharmaceutical Sciences, Zhengzhou University, Zhengzhou, China. [9]These authors contributed equally: Jie Zhong, Yuegui Guo, Shaoyong Lu. ✉e-mail: liuchenying@xinhuamed.com.cn; yangxy@shsmu.edu.cn; jian.zhang@sjtu.edu.cn

proteins, which caused the loss of the domains required for β-catenin binding but increased Asef activation through the APC-ARM domain. The Asef guanine nucleotide exchange factor (GEF) activity constitutively activated by APC-ARM binding promoted CRC cell migration via small Rho-like GTPase signaling[8–13]. These characteristics suggest that the APC–Asef PPI is a potential therapeutic target for metastatic CRC treatment; moreover, its inhibitors might be used as potential drugs to inhibit CRC cell migration.

To screen the peptide and peptidomimetic inhibitors of the APC–Asef PPI, a fluorescence-polarization (FP) competition assay[14,15] was developed in a previous study, wherein a fluorescein isothiocyanate (FITC)-labeled peptide served as a tracer (Ac–[176]GGG-GEQLAINELISDGK[FITC][195]–NH$_2$, tracer 1) in the FP assays. After screening a series of truncated peptides based on Asef residues 176–194, we found an inhibitor hit, MAI-005 ([181]GGEQLAI[187]), of the APC–Asef interaction PPI[16]. Then, we optimized MAI-005 into two peptidomimetic inhibitors, MAI-203 and MAI-400[16,17]. To aid in the translocation of peptidomimetic inhibitors across mammalian cell plasma membranes, we conjugated MAI-203 with a transcriptional transactivating (TAT) sequence and yielded MAIT-203, which inhibited CRC cell migration by disrupting the APC–Asef PPI[16]. However, the sensitivity of the FP assay was dependent upon the binding affinities of the FP tracers and target proteins[18–20]. FP tracers with limited affinities often exhibit unfavorable sensitivities[21,22], which markedly restricts their application in the screening of highly active PPI inhibitors[23,24]. For our APC–Asef PPI system, as the binding affinity of current tracer 1 was limited ($EC_{50} = 0.54 \pm 0.009$ μM), the FP assay could not be used to distinguish highly potent compounds. Moreover, this obstacle also occurred in other PPI inhibitor optimizations[25]. Thus, optimizing the tracer and increasing its binding affinity are crucial for improving FP assay sensitivity in the screening of PPI inhibitors.

In the present study, we devoted ourselves to improving the binding affinities of the tracers used in FP assays for the discovery of highly active inhibitors. By designing a series of tracers, we successfully obtained sensitivity-enhanced tracer 7 with a high binding affinity, markedly improving the resolution of FP assays for APC–Asef inhibitor screening. Moreover, using tracer 7, we discovered a best-in-class APC–Asef inhibitor, MAI-516, with an IC$_{50}$ of $0.041 \pm 0.004$ μM. We conjugated MAI-516 with the TAT sequence to obtain a cell-permeable inhibitor, MAIT-516, which could inhibit the migration of CRC cells with truncated APC by specifically hindering the APC–Asef PPI. Furthermore, MAIT-516 had no cytotoxic effects on the growth of normal intestinal epithelial cells and CRC cells expressing full-length or truncated APC. Overall, this sensitivity-enhancement strategy is suitable for screening PPI inhibitors, and MAIT-516 discovered by using this strategy would be an efficient APC–Asef inhibitor against CRC cell migration.

## Results

### Development of a sensitivity-enhanced tracer

In our previous study, we established an FP assay for screening APC–Asef inhibitors using FITC-labeled tracer 1 (Ac–[176]GGGGEQLAI-NELISDGK[FITC][195]–NH$_2$)[16]. With the help of tracer 1, the FP assay was used to distinctly quantify the activities of inhibitors with IC$_{50}$ values > 1 μM. In contrast, when the IC$_{50}$ values of the inhibitors were less than 1 μM, it was difficult to use the FP assay to distinguish between these values (Fig. 1a). For example, the IC$_{50}$ values of the previously discovered inhibitors MAI-150 (IC$_{50} = 1.1 \pm 0.06$ μM), MAI-203 ($0.57 \pm 0.06$ μM), and MAI-400 (IC$_{50} = 0.25 \pm 0.01$ μM) were shown to be within 2–4 folds of each other in the FP assay. However, their $K_d$ values measured by isothermal titration calorimetry (ITC) actually vary up to more than 20 times[16,17]. The results of the two methods indicated that the FP assay sensitivity is low for the evaluation of compounds with higher inhibiting potential, probably due to the lack of a good tracer, resulting in the weak ability to distinguish better PPI inhibitors

from others in large-scale optimization. Thus, developing a sensitivity-enhanced tracer may help improve the precision and measurement range of the FP assay for the identification of more potent inhibitors.

The tracer used in FP assays commonly contains three main subgroups: a peptide ligand with sufficient binding affinity for the target protein, a FITC subgroup with an appropriate quantum yield and fluorescence excited-state lifetime, and a linker between the ligand and the fluorophore. In most cases, the ligand subgroup was optimized to enhance the tracer binding affinity to improve the dynamic range of the FP assay. Recently, the FITC subgroup has been reported to contribute to the tracer binding affinity for some targets[26–29]. Although the FITC structure responsible for the fluorescence signal may not be modified, the linker subgroup conjugated to FITC can be optimized to regulate the binding mode of FITC to targets[30]. These findings indicated that both the ligand subgroup and the linker subgroup of the tracer could be optimized to improve the FP assay sensitivity.

To design a sensitivity-enhanced tracer, we constructed a computational model of the tracer 1–APC complex using molecular dynamics (MD) simulations (Fig. 1b). As shown in the model, the ligand subgroup of tracer 1 occupied the shallow pocket of APC, similar to Asef, while the FITC subgroup floated outside the tracer 1–APC interface due to the length and flexibility of the linker. First, we optimized the ligand subgroup of tracer 1 to improve its binding affinity. According to the tracer 1–APC model and our previous discovery of the corresponding structure-activity relationships (SAR)[12,16,17], more than 60% of the interactions between tracer 1 and APC are concentrated at a hotspot of "GGEQLAI". We modified the residues of the hot spot in tracer 1 and then obtained tracers 2 and 3 (Table 1, Fig. 1c). Compared to tracer 1, tracer 2 exhibited a two-fold increase in binding affinity ($EC_{50} = 0.18 \pm 0.003$ μM), and tracer 3 exhibited a ten-fold increase in binding affinity ($EC_{50} = 0.068 \pm 0.0009$ μM). Further replacement of the first residue in tracer 3 might introduce a hydrogen bonds or π–π stacking interactions with APC (Supplementary Fig. 1), yielding tracer 4 ($EC_{50} = 0.048 \pm 0.0006$ μM) and tracer 5 ($EC_{50} = 0.046 \pm 0.0006$ μM) with slight increases in potency and dynamic range. Moreover, we replaced the first two residues, "GG", of tracer 3 with a larger aromatic motif, a benzyloxycarbonyl (Z) group, and generated tracer 6 (Z–AGESLYENELISDGK[FITC]–NH$_2$, $EC_{50} = 0.040 \pm 0.0004$ μM) with a moderate increase in dynamic range.

Second, we optimized the linker subgroup to induce a suitable binding mode of the FITC subgroup to APC. To explore the effect of the linker, we synthesized a series of tracers with various linkers of different lengths and comprising different residues (Supplementary Fig. 2 and Supplementary Table 1). Finally, tracer 7 (Z–AGESLYEK[FITC]–NH$_2$, $EC_{50} = 0.016 \pm 0.0003$ μM) was found to show significant increases in both binding affinity and signal dynamic range ($\Delta mp = 306 \pm 2$ mp) (Fig. 1c, d). Our ITC measurements also confirmed the enhanced binding affinity of tracer 7 to APC ($K_d = 0.078$ μM), which was in good agreement with its EC$_{50}$ value determined by the FP assay (Fig. 2a, b, Supplementary Table 2). Collectively, through optimizing both the ligand and linker subgroups, we developed a sensitivity-enhanced tracer 7 with a 33-fold increase in EC$_{50}$ (Table 2) and a 25% increase in dynamic range (Table 1).

### Characterization of the tracer 7–APC complex structure

To reveal the binding mode of tracer 7 and APC, we determined the crystal structure of the tracer 7–APC complex with a resolution of 2.8 Å (Protein Data Bank (PDB) code: 7F7O, Supplementary Table 3, Supplementary Fig. 3a, c). As shown in Fig. 2c, tracer 7 is located on the pocket of APC-ARM and presents a binding mode similar to that of MAI-203 and MAI-400[16,17]. The A181 of tracer 7 (tracer 7-A181) forms two hydrogen bonds with the N594 and R549 of APC (APC-N594 and APC-R549), respectively, and tracer 7-G182 is fixed to APC-R549 through a hydrogen bond. Next, the side chain of tracer 7-E183 is stabilized through a salt bond with APC-K516, and the backbone of

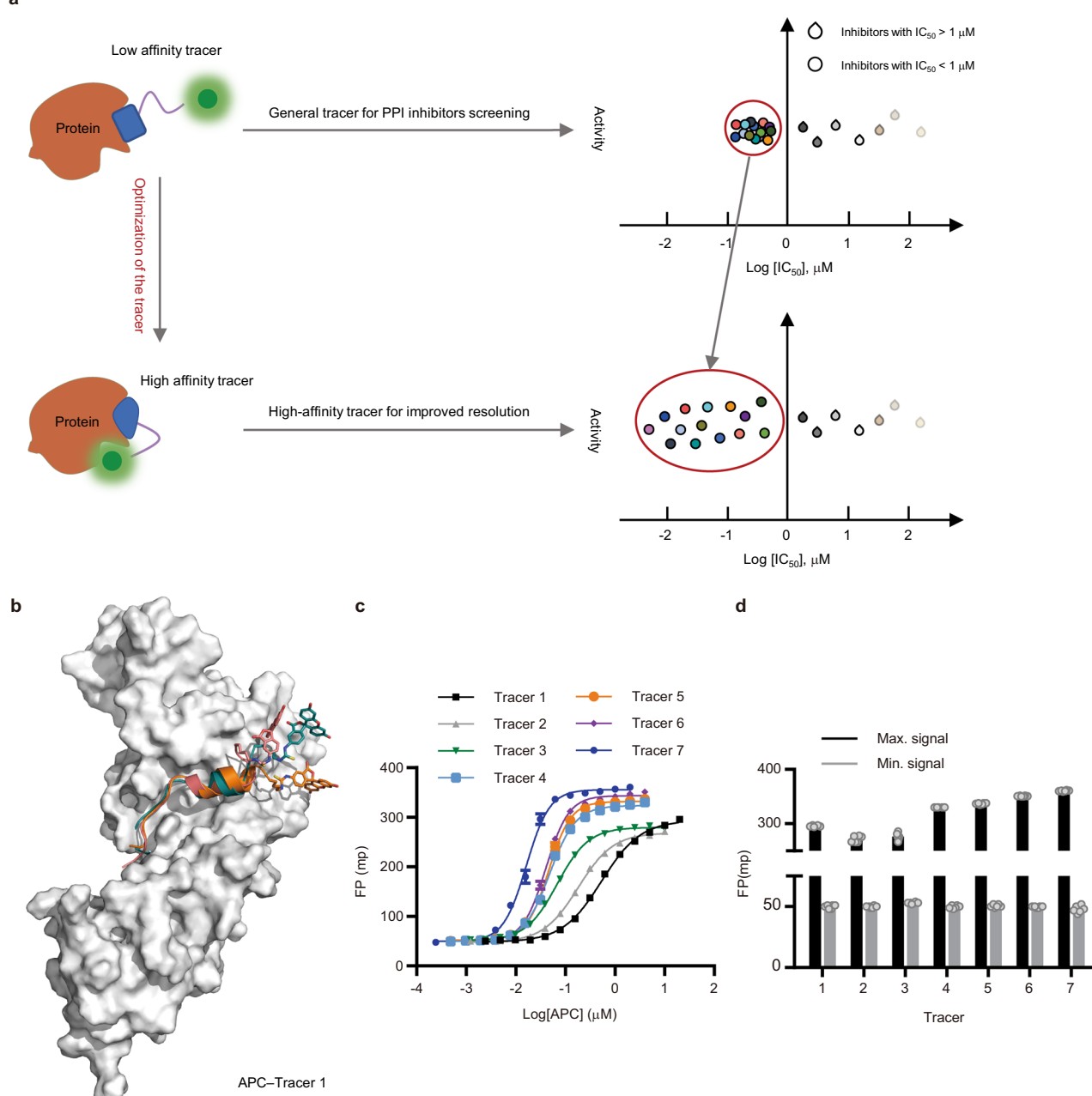

**Fig. 1 | Optimization of the FP tracer to improve the resolution of the FP assay.**
**a** Schematic illustration of the resolution of the APC−Asef inhibitor screening assay using tracer 1 and tracer 7. FP assays using tracer 1 were unable to be used to discriminate inhibitors with $IC_{50}$ values < 1 μM. FP assays using tracer 7 could be used to clearly distinguish inhibitors with $IC_{50}$ values < 1 μM. **b** Three significant binding conformations of tracer 1 to APC by molecular dynamics (MD) simulations. Pink: binding conformation of tracer 1 to APC at 36 ns; Blue: binding conformation of tracer 1 to APC at 110 ns; Orange: binding conformation of tracer 1 to APC at 340 ns. **c** Competitive binding curves for tracer 1 to tracer 7, determined by FP assays. Data are presented as the mean ± s.e.m ($n$ = 2 independent experiments). The experiments were performed in triplicates and repeated twice. **d** Maximum signal (black) and minimum signal (gray) of tracer 1 to tracer 7. The experiments were performed in triplicates and repeated twice ($n$ = 2 independent experiments).

tracer 7-S184 is within hydrogen-bonding distance to APC-N550. Moreover, the L185 and Y186 of tracer 7 form extensive van der Waals interactions with the F458, F510, M503, S546, and V543 of APC. Most remarkably, the FITC subgroup of tracer 7 occupies a hydrophobic area in the APC-ARM delimited by M454, L456, and F458, and the hydroxyl moiety of FITC interacts with APC-H462 via a hydrogen bond, demonstrating that FITC may serve as an additional binding anchor to its favored surface of APC when the linker subgroup between the ligand and FITC subgroups of tracer 7 is well optimized. This is consistent with our previously obtained biochemical/

biophysical results and provides insight into the structural basis for the high binding affinity of tracer 7.

## Improvement of sensitivity for FP assay with tracer 7

To test the sensitivity of tracer 7 in the FP assay, we re-evaluated the previous APC−Asef inhibitors using the fresh tracers. Compared to tracer 1, tracer 7 clearly distinguished inhibitors with $IC_{50}$ values <1 μM (Figs. 1a, 2d, Supplementary Table 4, 5) and reduced the APC protein concentration to 0.025 μM in the assay system (Table 2). Moreover, the $IC_{50}$ value obtained by tracer 7 revealed a good correlation with that of

**Table 1 | Binding affinities and dynamic ranges of tracers 1–7**

| Tracer | Sequence | EC$_{50}$ [µM][a] | Dynamic Range [Δmp] |
|---|---|---|---|
| 1 | Ac–GGGGEQLAINELISDGK[FITC]–NH$_2$ | 0.54 ± 0.009 | 246 ± 1.6 |
| 2 | Ac–GGAGEALAINELISDGK[FITC]–NH$_2$ | 0.18 ± 0.003 | 221 ± 1.4 |
| 3 | Ac–GGAGESLYENELISDGK[FITC]–NH$_2$ | 0.068 ± 0.0009 | 229 ± 1.1 |
| 4 | Ac–YGAGESLYENELISDGK[FITC]–NH$_2$ | 0.048 ± 0.0006 | 274 ± 1.4 |
| 5 | Ac–EGAGESLYENELISDGK[FITC]–NH$_2$ | 0.046 ± 0.0006 | 280 ± 1.5 |
| 6 | Z–AGESLYENELISDGK[FITC]–NH$_2$ | 0.040 ± 0.0004 | 294 ± 1.3 |
| 7 | Z–AGESLYEK[FITC]–NH$_2$ | 0.016 ± 0.0003 | 306 ± 2.0 |

[a]The effects of the tracers in the fluorescence polarization (FP) competition assay were assessed as described in the Experimental Section. Data are presented as the mean ± s.e.m ($n$ = 2 independent experiments). The experiments were performed in triplicates and repeated twice. Source data are provided as a Source Data file.

tracer 1 (IC$_{50}$ > 1 µM) (Fig. 2e). Using tracer 7 for the FP assay, we obtained a higher $Z'$ factor of 0.92, indicating a high sensitivity and resolution assay for screening application (Table 2). Thus, the subsequent application of sensitivity-enhanced tracer 7 in FP assays will allow the optimization of more potent APC–Asef inhibitors.

## Discovery of APC–Asef inhibitor MAI-516 using FP assay with tracer 7

To better design APC–Asef inhibitors, we analyzed the structures of the MAI-203−APC and MAI-400−APC complexes. As shown in Supplementary Fig. 4, the distance between the benzyl moiety of MAI-400 and the indoyl moiety of APC-W593 was approximately 4.8–5.7 Å. We hypothesized that the enhancement of the π-π interaction at position R$^1$ (Fig. 3a and Table 3) might increase the binding affinity of the inhibitor to APC. In addition, the hydrophobic effects of the R$^2$ and R$^3$ substituents of MAI-203 and MAI-400 could be optimized based on the surrounding APC hydrophobic residues (Table 4). Then, a series of peptidomimetics (MAI-501 to MAI-522) were synthesized and evaluated using the FP assay with tracer 7.

To explore the enhancement of the π-π interaction at the R$^1$ position, MAI-511 with a phenyl substituent was designed and showed a significant increase in binding affinity (IC$_{50}$ = 0.098 ± 0.012 µM) compared to MAI-203 and MAI-400 (Fig. 3b, Table 3, and Supplementary Fig. 5). In contrast, MAI-501−MAI-505 with aliphatic substituents at the R$^1$ position exhibited decreased binding affinities relative to that of MAI-511, indicating that the aromatic group may play a positive role in the enhancement of the π-π interaction, which was in agreement with our structural analysis. Next, we investigated the effect of substituting heterocyclic rings at the R$^1$ position. As shown in Table 3, the analogs with five-membered aromatic ring, including furan (MAI-506, IC$_{50}$ = 0.40 ± 0.07 µM), thiophene (MAI-507, IC$_{50}$ = 0.17 ± 0.007 µM), thiazole (MAI-508, IC$_{50}$ = 0.22 ± 0.02 µM), pyrrole (MAI-509, IC$_{50}$ = 0.57 ± 0.04 µM), and methyl-pyrrole (MAI-510, IC$_{50}$ = 0.34 ± 0.03 µM), at the R$^1$ position exhibited two- to six-fold decreases in binding affinity compared to MAI-511. Similarly, substituting peptidomimetics with six-membered heterocyclic rings, such as pyridine (MAI-512 to MAI-514), at the R$^1$ position also failed to improve the binding affinity of APC. These results suggested that the presence of polar atoms in aromatic rings might weaken the π-π interactions between the local residues of the APC pocket. Next, we modified the substituents on the phenyl ring at the R$^1$ position based on the structure of MAI-511, leading to the improved inhibitor MAI-516 with an IC$_{50}$ of 0.041 ± 0.004 µM (Table 3). We also modified the substituents at the R$^2$ (e.g., MAI-517−MAI-520) and R$^3$ positions (e.g., MAI-521−MAI-522) by changing their sizes and constrained conformations, and the binding affinities of most of the resulting analogs were maintained (Table 4). According to these aforementioned results, MAI-516 was selected for application in subsequent pharmacological evaluation.

The binding affinity of MAI-516 to APC was confirmed by surface plasmon resonance (SPR) and ITC. The SPR experiments indicated that MAI-516 reversibly interacted with APC in a dose-dependent manner (Fig. 3c–e). As summarized in Table 5, the obtained affinity constant $K_{on}$ was [4.0 ± 0.4] × 10$^5$ (M$^{-1}$ s$^{-1}$), and the off-rate constant $K_{off}$ was [7.4 ± 1.5] × 10$^{-3}$ (s$^{-1}$). The equilibrium dissociation constant value of MAI-516 ($K_D$ = 0.018 ± 0.002 µM) was derived from the ratio between the kinetic dissociation ($K_{off}$) and association ($K_{on}$) constants obtained by fitting the data from 1.56 to 200 nM using the simple 1:1 Langmuir binding fit model. As shown in the affinity fitting plots, MAI-516 had a better kinetic $K_D$ value for binding to APC than MAI-203, which was in good agreement with FP assay results (Fig. 3e). Moreover, the ITC assay results confirmed that MAI-516 tightly interacted with APC with a $K_d$ value of 4.4 nM (Fig. 3f and Table 6), which was consistent with the FP and SPR assay results.

## Structural analysis of the MAI-516−APC complex

To elucidate the structural basis for the binding of MAI-516 to APC, we determined the crystal structure of the MAI-516−APC complex (PDB code: 7F6M, Supplementary Table 3). The overall structure revealed an apparent electron density of MAI-516 bound in the APC-ARM pocket with a resolution of 2.7 Å (Supplementary Fig. 6a, Supplementary Fig. 3b, d). As shown in Fig. 3g, MAI-516 induces the conformational rearrangement of APC-R549 to form a druggable doughnut pocket in the APC hotspot. Proximal to the N-terminus of MAI-516, the A181 of MAI-516 (MAI-516-A181) forms respective hydrogen bonds with APC-N594 and APC-R549, and MAI-516-G182 forms a hydrogen bond with APC-R549. The carboxylate moiety of MAI-516-E183 accepts a charge-stabilized hydrogen bond from the amino moiety of APC-K516 and creates a pair of hydrogen bonds with the main chain of APC-G511. Meanwhile, the backbone of MAI-516-S184 forms two hydrogen bonds with the side chain of APC-N550. At the C-terminus of MAI-516, MAI-516-L185 and MAI-516-Y186 cluster and create extensive van der Waals interactions with APC-F458, APC-F510, APC-M503, APC-S546, and APC-V543 (Supplementary Fig. 6b). In addition, MAI-516-Y186 forms a hydrogen bond with APC-Q542.

Consistent with our modeling prediction, the N-terminal 4-methoxy-phenyl forms parallel-displaced π−π stacking with the indoyl of APC-W593 with a distance of 3.7–4.6 Å, thereby allowing the methoxy moiety of MAI-516 to form a hydrogen bond with the NH$_2$ of APC-N641 (Fig. 3g, Supplementary Fig. 6b). These results suggest that the newly formed π-π stacking interaction and hydrogen bond are crucial for increasing the binding affinity of MAI-516 (Supplementary Fig. 7). To confirm the π-π stacking interaction, we respectively mutated APC-W593 and APC-N641 to alanine and determined the binding affinity of MAI-516 to the resulting APC mutants via ITC assays (Table 6). The binding affinity of MAI-516 to the W593A and N641A APC mutants was reduced by approximately 100-fold (Table 6, Supplementary Fig. 8a) and 2-fold (Table 6, Supplementary Fig. 8b), respectively. The binding affinity of MAI-516 to the W593A and N641A APC double-mutant was almost eliminated (Table 6, Supplementary Fig. 8c) with a 200-fold reduction. Overall, these results indicated that the newly formed π-π stacking interaction and hydrogen bond were the primary determinants for the improved activity of MAI-516.

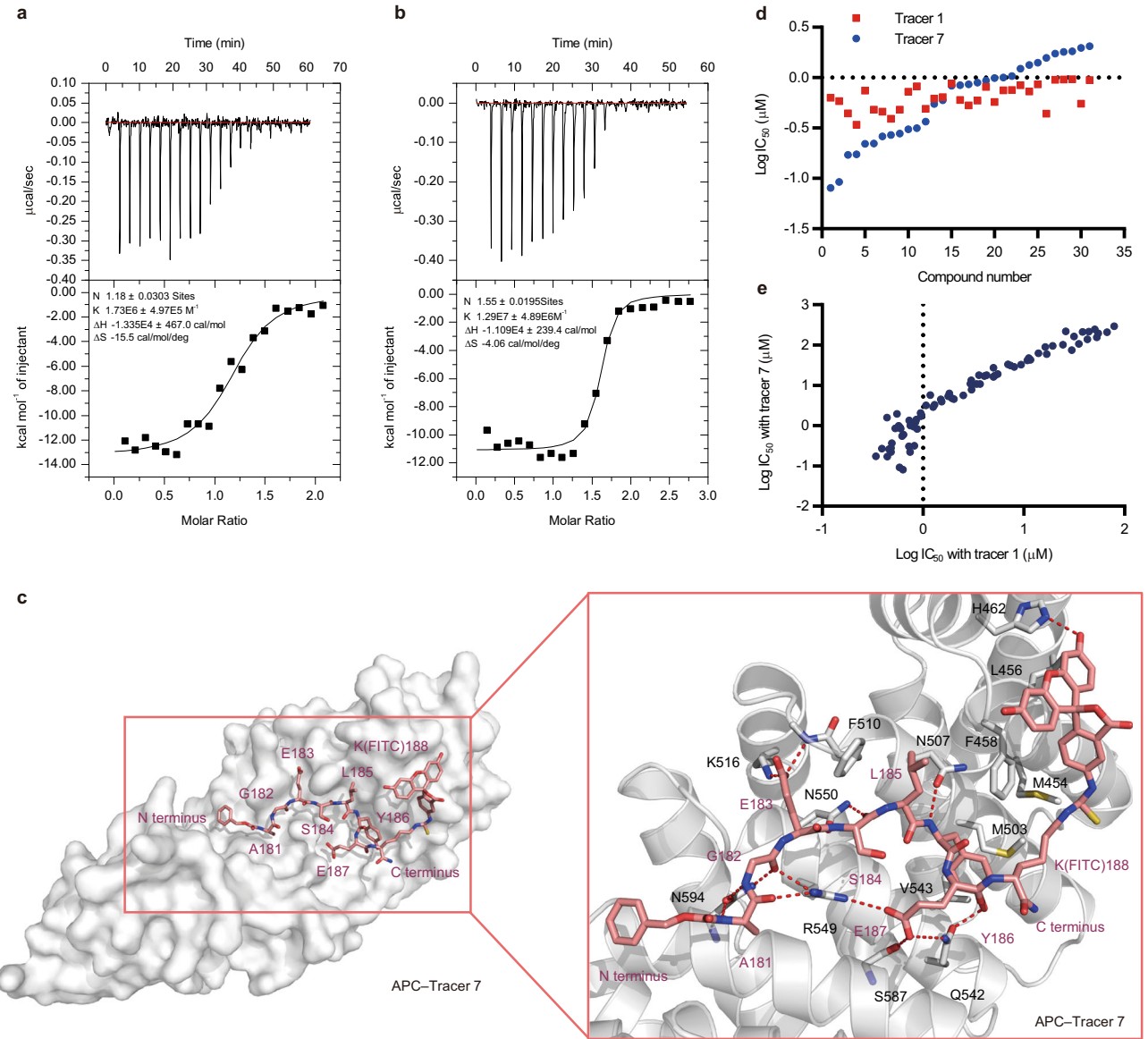

**Fig. 2 | Comparison of tracer 1 and tracer 7 in terms of binding affinity, binding mode and resolution of the FP assay. a** Isothermal titration calorimetry (ITC) experiments for the binding of tracer 1 to APC. $N$: the number of sites per APC, $K$: the binding constant between APC and tracer 1, $\Delta H$: heat change, $\Delta S$: entropy change. The data shown are representative of three independent experiments. The upper plot presents the ITC raw data, and the lower plot presents fitted integrated peak areas. **b** ITC experiments for the binding of tracer 7 to APC. **c** Characterization and cocrystal structure of the tracer 7–APC complex (PDB code: 7F7O). APC is shown as a solvent-accessible surface (gray), and tracer 7 is depicted by sticks (pink carbon atoms). The red dashed lines represent the hydrogen bonds between tracer 7 (pink carbon atoms) and APC (gray carbon atoms). **d** Log-transformed half-maximal inhibitory concentration (log($IC_{50}$)) of 31 APC–Asef PPI inhibitors was determined by FP assays with tracer 1 (red) and tracer 7 (blue). **e** The correlation of log-transformed half-maximal inhibitory concentration (log($IC_{50}$)) of 80 APC–Asef PPI inhibitors determined by FP assays with tracer 1 and tracer 7. Inhibitors with half-maximal inhibitory concentrations ($IC_{50}$) less than 1 µM under tracer 1 are better distinguished by tracer 7.

**Table 2 | Comparison of tracer 1 and tracer 7**

| Tracer | EC$_{50}$ [µM] | $K_d$ [µM] | Protein conc. [µM] | Tracer conc. [nM] | Span [mp] | Z' |
|---|---|---|---|---|---|---|
| 1 | 0.54 ± 0.009 | 0.58 | 1.0 | 20 | 134 | 0.83 |
| 7 | 0.016 ± 0.0003 | 0.078 | 0.025 | 20 | 251 | 0.92 |

The experiments were performed in triplicates and repeated twice (n = 2 independent experiments). Source data are provided as a Source Data file.

## Disruption of the APC–Asef PPI by MAI-516 in cell models

To investigate the effect of MAI-516 on the APC–Asef PPI, we first examined whether MAI-516 could perturb the APC–Asef PPI in cell lysates. HEK293T cells were transiently transfected with FLAG-tagged APC (303–876) and hemagglutinin (HA)-tagged Asef (170–632) plasmids. The cell lysates were harvested several hours after transfection and treated with different concentrations of MAI-203 and MAI-516 for

2 h; then, they were incubated with anti-FLAG® M2 affinity gel (Sigma; A2220) for 3 h at 4 °C. We observed that MAI-516 could dose-dependently attenuate the APC–Asef PPI in cell lysates, and that this attenuation was more effective than that of MAI-203 (Supplementary Fig. 9a).

As previously reported[16], MAI-516 has difficulty crossing cell membranes to reach intracellular targets. To increase the permeability

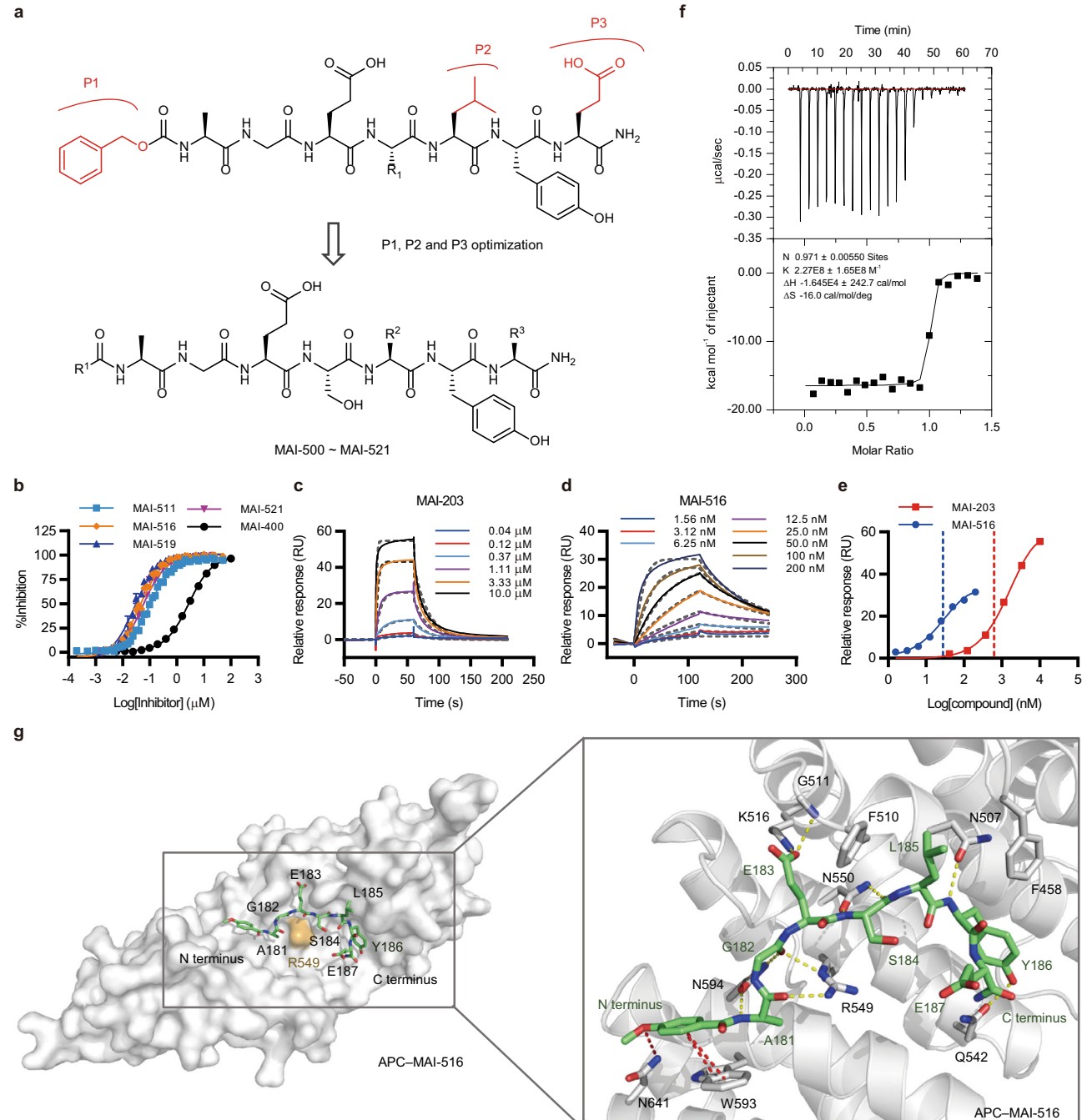

**Fig. 3 | Design, biochemical and structural characterization of high-affinity APC–Asef inhibitors with tracer 7. a** The optimization strategy of APC–Asef inhibitors based on the structure of MAI-400. P1, P2, and P3 (red) were modified by the rational optimization strategy. **b** Competitive binding curves for MAI-511, MAI-516, MAI-519, MAI-521, and MAI-400, determined by FP assay. Data are presented as the mean ± SD; $n = 3$ independent experiments. **c, d** Representative SPR sensorgrams (solid lines) and corresponding Langmuir 1:1 kinetic model fit (dashed lines) for MAI-203 **c** and MAI-516 **d** binding to APC. **e** Representative plot of the response

signals at equilibrium against the concentrations of MAI-203 (red) and MAI-516 (blue). **f** ITC experiments for the binding of MAI-516 to APC. **g** Characterization and cocrystal structure of the MAI-516–APC complex (PDB code: 7F6M). APC is shown as a solvent-accessible surface (gray), and MAI-516 is depicted by sticks (green carbon atoms). The red dashed lines represent the newly formed hydrogen bonds and π-π stacking interactions between MAI-516 (green carbon atoms) and APC (gray carbon atoms). The yellow dashed lines represent the hydrogen bonds between MAI-516 and APC.

of MAI-516, we conjugated the C-terminus of MAI-516 with an optimized "GGGGG" linker and cell-penetrating peptides (TAT) and acquired the cell-permeable inhibitor MAIT-516. To assess the potential effect of this TAT + linker addition on APC–Asef complex disruption, we used an ITC assay to measure the binding affinities of MAI-516, TAT + linker, and MAIT-516 to APC. The results showed that there was no apparent binding between TAT + linker and APC, and the

dissociation constant $K_d$ values of MAI-516 and MAIT-516 were 4.4 nM and 67 nM, respectively (Fig. 3f, Supplementary Fig. 8d, e, and Table 6), demonstrating that the TAT + linker is solely responsible for cell penetration.

We next performed the coimmunoprecipitation (Co-IP) of exogenous HA–APC (303–876) and FLAG–Asef (170–632) in HEK293T cells treated with MAI-516, TAT + linker, and MAIT-516.

**Table 3 | Peptide inhibitor MAI-501–MAI-516 structures and APC binding affinities as measured by FP-based assays**

| Compound | R¹ | IC₅₀ ± SD [µM]ᵃ | Compound | R¹ | IC₅₀ ± SD [µM]ᵃ |
|---|---|---|---|---|---|
| MAI-501 |  | 24 ± 6 | MAI-510 |  | 0.34 ± 0.03 |
| MAI-502 |  | 2.1 ± 0.5 | MAI-511 |  | 0.098 ± 0.012 |
| MAI-503 |  | 5.8 ± 1.1 | MAI-512 |  | 0.98 ± 0.06 |
| MAI-504 |  | 1.8 ± 0.4 | MAI-513 |  | 0.29 ± 0.05 |
| MAI-505 |  | 1.6 ± 0.09 | MAI-514 |  | 0.11 ± 0.014 |
| MAI-506 |  | 0.40 ± 0.07 | MAI-515 |  | 0.24 ± 0.03 |
| MAI-507 |  | 0.17 ± 0.007 | MAI-516 |  | 0.041 ± 0.004 |
| MAI-508 |  | 0.22 ± 0.02 | MAI-400 |  | 2.8 ± 0.05 |
| MAI-509 |  | 0.57 ± 0.04 | | | |

ᵃ The effects of the APC peptide inhibitors in the fluorescence-polarization (FP) competition assay were assessed as described in the Experimental Section. The IC₅₀ values shown are the averages of three independent experiments with a typical variation of less than 20%.

$n = 3$ independent experiments. Source data are provided as a Source Data file.

Although MAI-516 could efficiently attenuate the APC–Asef PPI in the cell lysate, the results showed that MAI-516 and TAT + linker (50 µM) had no effect on the APC–Asef PPI in cell models (Fig. 4a). However, MAIT-516 significantly diminished the APC–Asef PPI in a dose-dependent manner, presenting an improvement in the cell membrane permeability of MAIT-516 by the TAT group (Fig. 4a). Moreover, MAIT-516 was more potent at blocking the APC – Asef PPI in HEK293T cells than MAIT-203 at a concentration of 50 µM (Fig. 4a).

We further performed the endogenous Co-IP of APC–Asef in response to MAIT-516 in CRC cells expressing truncated APC (SW480, DLD1) and WT full-length APC (RKO). Consistent with the results of overexpressed tagged proteins in HEK293T cells, the interaction between the endogenous truncated APC and Asef was dramatically diminished by MAIT-516 treatment in both the SW480 and DLD1 cells (Fig. 4b, Supplementary Fig. 9b). In contrast, MAIT-516 showed little effect on the endogenous PPI between WT APC and Asef (Supplementary Fig. 9c), which demonstrated the specific inhibition of the truncated APC–Asef PPI by MAIT-516.

## Inhibition of the migration of CRC cells with truncated APC expression by MAIT-516

To explore the effect of MAIT-516 on APC–Asef-mediated CRC cell migration, we accomplished a wound-healing assay. The results revealed that MAIT-516 could suppress SW480 cell migration at 5 µM (Fig. 4c, d), which is better than MAIT-203, consistent with their binding affinities to APC in vitro. We also employed the xCELLigence Real-Time Cell Analysis (RTCA) system to quantitatively measure cell migration in real time. The results exhibited that MAIT-516 reduced the migration of SW480 cells in a dose-dependent manner. Moreover, the anti-migration effect of MAIT-516 at 10 µM was more potent than that of MAIT-203, which was in agreement with the quantitative binding affinities of the inhibitors to APC (Fig. 4e). In transwell-based migration assays, compared with DMSO and MAIT-203, MAIT-516 also more significantly decreased the migration of SW480 cells ($P < 0.001$, Fig. 4f, g). In addition, we observed that MAI-516 had no apparent impact on the migration of SW480 cells in wound-healing, transwell, and RTCA migration assays (Supplementary Fig. 10), whereas MAIT-516 significantly inhibited the migration of SW480

**Table 4 | Peptide inhibitor MAI-517–MAI-522 structures and APC binding affinities as measured by FP-based assays**

| Compound | R2 | R3 | IC50 ± SD [μM][a] |
|---|---|---|---|
| MAI-517 | | | 0.025 ± 0.004 |
| MAI-518 | | | 0.045 ± 0.002 |
| MAI-519 | | | 0.025 ± 0.004 |
| MAI-520 | | | 0.051 ± 0.007 |
| MAI-521 | | | 0.053 ± 0.0013 |
| MAI-522 | | | 0.15 ± 0.017 |
| MAI-400 | | | 2.8 ± 0.05 |

[a] The effects of the APC peptide inhibitors in the fluorescence-polarization (FP) competition assay were assessed as described in the Experimental Section. The IC50 values shown are the averages of three independent experiments with a typical variation of less than 20%.
n = 3 independent experiments. Source data are provided as a Source Data file.

**Table 5 | Kinetic parameters of MAI-203 and MAI-516 binding to APC as measured by SPR experiments**

| Compound | $k_{on}$ [M⁻¹ s⁻¹][a] | $k_{off}$ [s⁻¹][b] | RT [s][c] | $K_D$ [μM][d] |
|---|---|---|---|---|
| MAI-203 | $[7.8 ± 0.2] × 10^4$ | $[4.4 ± 0.5] × 10^{-2}$ | 23 | 0.57 ± 0.08 |
| MAI-516 | $[4.0 ± 0.4] × 10^5$ | $[7.4 ± 1.5] × 10^{-3}$ | 135 | 0.018 ± 0.002 |

[a]$k_{on}$ ± SD. [b]$k_{off}$ ± SD. [c]RT = 1/$k_{off}$, RT is expressed in seconds. [d]$K_D$ = $k_{off}$/$k_{on}$. n = 3 independent experiments. Source data are provided as a Source Data file.

**Table 6 | Thermodynamic data for MAI-516 binding to APC and mutated APC**

| Compound | Protein | L/P ratio | $K_d$ [μM] | ΔG [kJ mol⁻¹] | ΔH [kJ mol⁻¹] | -TΔS [kJ mol⁻¹][a] |
|---|---|---|---|---|---|---|
| MAI-516 | APC WT | 0.97 | 0.0044 | −48.5 | −68.8 | 20.3 |
| MAI-516 | APC W593A | 0.92 | 0.40 | −37.1 | −48.1 | 11.0 |
| MAI-516 | APC N641A | 0.88 | 0.012 | −46.0 | −68.8 | 22.8 |
| MAI-516 | APC W593A, N641A | 0.77 | 0.86 | −35.3 | −55.8 | 20.5 |
| MAIT-516 | APC WT | 1.6 | 0.067 | −41.7 | −36.8 | −4.90 |
| TAT + linker | APC WT | N.A. | N.A. | N.A. | N.A. | N.A. |

[a]All experiments were performed at 30 °C. The L/P ratio indicates the number of sites per APC. ΔH: change in enthalpy; -TΔS: change in entropy; $K_d$: equilibrium dissociation constant determined by ITC.

cells (Fig. 4c–g), which was consistent with the results from Co-IP results (Fig. 4a) and further demonstrated that the TAT group improved the cell membrane permeability of MAIT-516.

To further investigate the specificity of the inhibitory effect of MAIT-516 on CRC cell migration, we performed wound healing and transwell migration assays with normal intestinal epithelial cells and CRC cells expressing full-length APC or truncated APC. Our results showed that MAIT-516 had no noticeable influence on the migration of either normal intestinal epithelial cells (HIEC-6) (Supplementary Fig. 11) or CRC cells expressing full-length APC (LS-1034, RKO, SNU-C2B) (Supplementary Fig. 12). In contrast, MAIT-516 significantly restrained the migration of CRC cells expressing truncated APC (Caco-2, DLD-1, HT-29, LoVo) at 25 μM (Supplementary Fig. 13) in addition to reducing the migration of SW480 cells. These data indicated that the suppressive effect of MAIT-516 on cell migration was specific to CRC cells with truncated APC expression.

As the W593A and N641A mutants significantly reduced the binding affinity of MAI-516 to truncated APC, to inspect the biological relevance of these mutants in CRC cells, we constructed SW480 stable cells overexpressing truncated APC (1–1338, the same as the endogenous APC expressed in SW480 cells) and the W593A and N641A mutants (Supplementary Fig. 14a). Along with the previous reports[8], the overexpression of truncated APC dramatically enhanced the migration of SW480 cells (Supplementary Fig. 14b, c). Moreover, the W593A mutant almost abrogated the enhanced cell migration induced

by overexpression of truncated APC, while the N641A mutant diminished the promigratory activity of truncated APC to a lesser extent (Supplementary Fig. 14b, c). In addition, MAIT-516 significantly reduced the migration of SW480 cells overexpressing truncated WT APC but not that of cells overexpressing W593A or N641A mutants (Supplementary Fig. 14b, c). These data further supported the vital role of W593/N641 in increasing the binding affinity of MAI-516 to APC.

### MAIT-516 shows no impact on cell proliferation, apoptosis, or differentiation in colorectal epithelial cells

The activity of β-catenin is essential for intestinal homeostasis and the survival of major CRC cells. Thus, we explored whether MAIT-516 could influence canonical Wnt signaling. The protein half-life and the subcellular localization of β-catenin in SW480 cells treated with MAIT-516 were similar to those in mock-treated SW480 cells (Supplementary Fig. 15a–d). Consistently, quantitative polymerase chain reaction (qPCR) analysis revealed that the mRNA levels of β-catenin target genes (AXIN2, LGR5, CCND1, MYC) were not affected by MAIT-516 treatment in SW480 cells (Supplementary Fig. 15e–h). These data suggested that diminishing the APC–Asef PPI did not affect canonical Wnt signaling in CRC cells with truncated APC expression.

We next estimated the general cytotoxicity of MAIT-516 in normal intestinal epithelial cells (HIEC-6) and CRC cells expressing full-length (RKO, LS-513, LS-1034, NCI-H508, NCI-H716, and SNU-C2B) or truncated (Caco-2, HT-29, LoVo, DLD-1, SW480, and SW620) APC using the CCK-8 assay[16,27]. Our results showed that both MAIT-203 and MAIT-516 at concentrations up to 50 μM did not impair the viability of normal intestinal epithelial cells and CRC cells, regardless of which type of APC was expressed (Supplementary Fig. 16a). Furthermore, we assessed the toxicity of MAIT-516 in primary organoids derived from mouse intestines and paired human normal and colorectal tissues. Consistently, MAIT-516 showed little cytotoxicity in these primary organoid models (Supplementary Fig. 16b–f). Second, we performed cell apoptosis assays with HIEC-6 normal intestinal cells and SW480 CRC cells treated with MAIT-516. In agreement with the CCK8 assay results, MAIT-516 did not induce the apoptosis of either HIEC-6 or SW480 cells (Supplementary Fig. 17a, b). These data

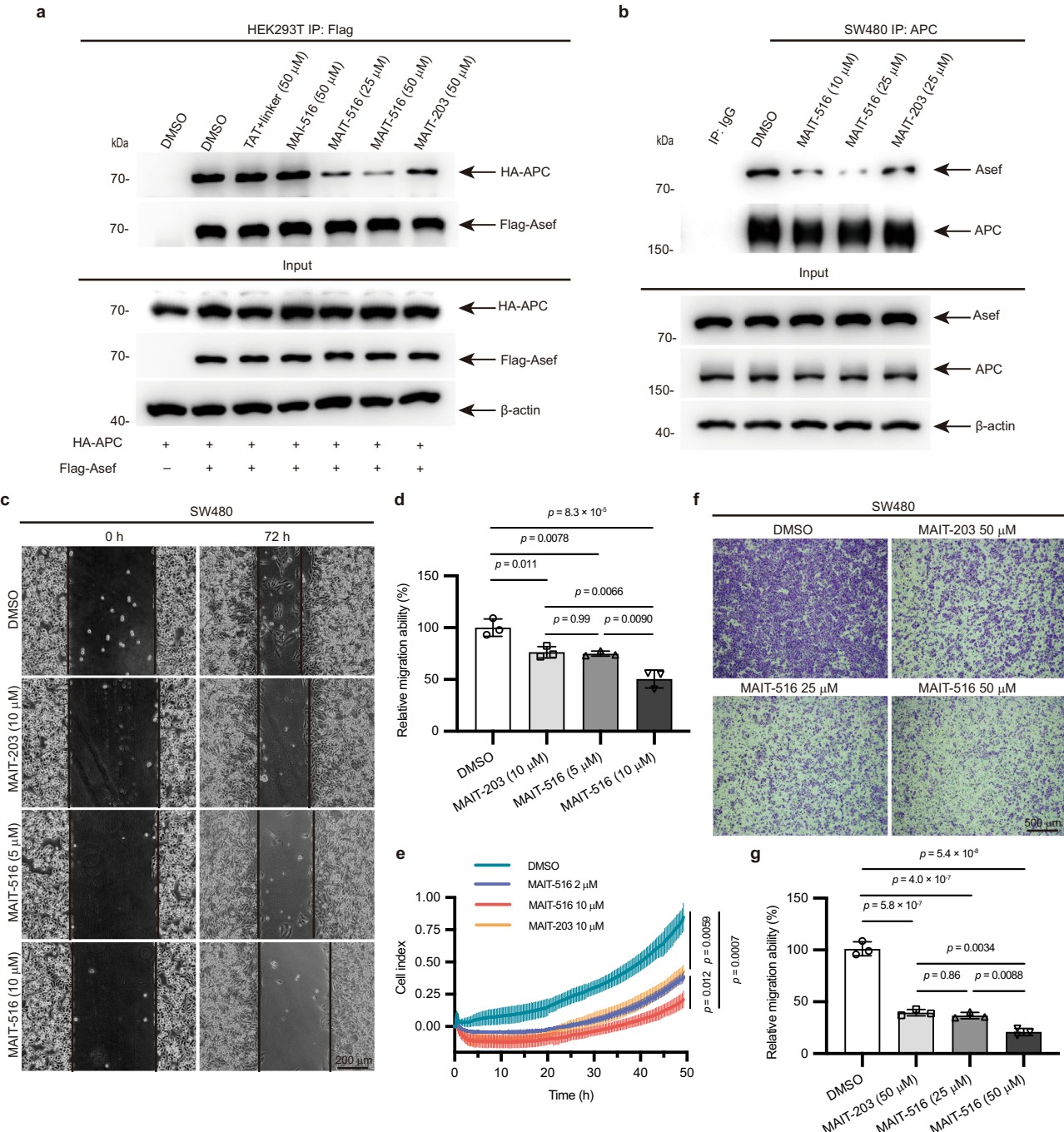

**Fig. 4 | MAIT-516 inhibits the migration of SW480 cells by disrupting the APC–Asef PPI. a** Co-IP of exogenous HA–APC (303–876) and Flag–Asef (170–632) in HEK293T cells treated with DMSO, TAT + linker (50 μM), MAI-516 (50 μM), MAIT-516 (25 μM and 50 μM), and MAIT-203 (50 μM) for 24 h. **b** Western blot analysis for Co-IP with anti-APC antibody in SW480 cells treated with 10 μM or 25 μM MAIT-516. Full blots are provided in Source Data files. All co-IP experiments were repeated three times with similar results. **c**, **d** Representative images **c** of the wound-healing assays of SW480 cells treated with DMSO, MAIT-516, or MAIT-203 for 72 h. The images shown are representative of triplicate experiments. Scale bar: 20 μm. Quantitative cell migration data **d** are expressed relative to the migration ability of the DMSO-treated group. Data are represented as the mean ± SD (n = 3 independent experiments). **e** Kinetic curves of the migration of SW480 cells after treatment with DMSO, MAIT-516 or MAIT-203 as assessed by xCELLigence RTCA-DP. Data are presented as the mean ± SD (n = 4). The experiments were performed in 4 replicates and repeated three times. p values were calculated by two-way ANOVA with Tukey's multiple comparisons test (DMSO vs. MAIT-516 2 μM: p = 0.0059; DMSO vs. MAIT-203 10 μM: p = 0.0059; DMSO vs. MAIT-516 10 μM: p = 0.0007; MAIT-516 2 μM vs. MAIT-203 10 μM: p = 0.66; MAIT-516 2 μM vs. MAIT-516 10 μM: p = 0.027; MAIT-203 10 μM vs. MAIT-516 10 μM: p = 0.012). **f**, **g** Representative images **f** of the transwell migration assay of SW480 cells treated with DMSO, MAIT-516, or MAIT-203. The images shown are representative of triplicate experiments. Scale bar: 500 μm. The number of cells that migrated was normalized to the average cell count of DMSO-treated cells **g**. Data are shown as the mean ± SD (n = 3 independent experiments). p values were calculated by one-way ANOVA with Tukey's multiple comparisons test **d**, **g**. Source data are provided as a Source Data file.

indicated the low cytotoxicity of MAIT-516 for both normal and cancer intestinal tissues.

Finally, we evaluated the potential effect of MAIT-516 on intestinal cell differentiation. Notch inhibition by dibenzazepine (DBZ) allows colorectal stem cells to differentiate into goblet cells[28], and the DBZ-induced goblet cell differentiation of LS-174T cells is a well-established intestinal differentiation model[29]. We observed that NOTCH inhibition by DBZ significantly suppressed the mRNA level of the NOTCH target gene *HES1*, while increasing the mRNA levels of goblet cell markers (*MUC2, TFF3, ATOH1, SPDEF*), which indicated induced goblet cell differentiation (Supplementary Fig. 17c). In addition, we found that MAIT-516 treatment did not affect the gene expression of the goblet cell markers, which implied that MAIT-516 had no impact on intestinal cell differentiation (Supplementary Fig. 17c).

Collectively, the above results elucidated that MAIT-516 with improved activity compared to MAIT-203 specifically inhibited the metastasis of CRC cells expressing truncated APC by the suppression of APC–Asef-mediated cell migration. The low cytotoxicity and cancer-specific effect of MAIT-516 suggest its potential in future drug development.

## Discussion

Our study designed a sensitivity-enhanced tracer 7 that can be used in FP assays to quantify high-affinity APC–Asef inhibitors. Using tracer 7, we discovered a best-in-class APC–Asef inhibitor, MAI-516 (IC$_{50}$ = 0.041 ± 0.004 μM), which efficiently blocked the APC–Asef PPI in cell lysates in a dose-dependent manner. By conjugating the C-terminus of MAI-516 with an optimized "GGGGG" linker and TAT, we obtained the cell-permeable inhibitor MAIT-516. MAIT-516 specifically inhibited the metastasis of CRC cells expressing truncated APC by suppressing APC–Asef-mediated cell migration rather than influencing the canonical Wnt signaling pathway. Moreover, MAIT-516 had little effect on cell proliferation, apoptosis, and differentiation. Our present work demonstrated the feasibility of a sensitivity-enhanced tracer in FP assays for APC–Asef inhibitor screening and validated the effective inhibitor MAIT-516 as a potential candidate in future drug development for CRC therapy.

PPIs present a promising class of drug targets, and there are a variety of biochemical assays for PPI inhibitor screening, including fluorescent resonance energy transfer (FRET), AlphaScreen, enzyme-linked immunosorbent (ELISA), and FP assays. FRET and AlphaScreen assays are proximity measurements that rely on protein partners (within 10–100 Å). However, the FRET assay costly, and the signal window for FRET experiments depends on distance and dye orientation[30–32]. The AlphaScreen assay can quantify binding interactions even for very large proteins and phage particles due to the large diffusion distance of singlet oxygen, whereas the limitations of the AlphaScreen assay are high cost and material sensitivity to ambient light[33–35]. The ELISA assay is flexible and sensitive but has multiple incubation and washing steps, which are time-consuming for automated and bench-top assays[36–38]. Compared to these methods, the FP assay is a sensitive nonradioactive method and is an in homogenous "mix-and-read" format without washing steps. In addition, the FP assay is directly carried out in solution; no perturbation of the sample is required, resulting in measurements that are faster and more native-like than those of the ELISA assay. Furthermore, the FP assay is readily adaptable to low volumes. Due to its technical simplicity and low cost, the FP assay has been widely used to primarily screen PPI inhibitors[39–41], and the development of a highly sensitive FP assay with a well-designed tracer would provide an avenue for supporting the optimization of more potent inhibitors.

In our previous studies, we established an FP competitive assay to screen APC–Asef inhibitors[16]. The sensitivity of the FP assay was dependent upon the binding affinities of FP tracers and target proteins. However, FP tracers with limited affinities often exhibit unfavorable sensitivities, which markedly restricts their application in screening PPI inhibitors. To discover potent APC–Asef inhibitors, the origin tracer is insufficient for high resolution and sensitivity. Jan *et al.* showed that the addition of FITC improved the binding affinities of their inhibitors to glutamate carboxypeptidase II[26]. Vincenzo et al. revealed that varying the length of the linker between FITC and the inhibitor sulfonamide affected its inhibition activities against carbonic anhydrase[42]. These findings indicate that as a moiety of the tracer, FITC not only offers fluorescence for real-time detection but also increases the binding affinity of the tracer to the target, resulting in an improved FP assay sensitivity. Herein, we proposed that the coattachment of the ligand and fluorophore to the protein might improve the binding affinity of the tracer to the protein. Based on this strategy, we modified the distance between FITC and the peptide to adjust the binding site of APC by introducing a hydrogen bond between FITC and APC. This interaction contributed to the increased binding affinity of tracer 7 to APC and precluded the propeller effect. The increased binding affinity of the tracer significantly improved the dynamic range, sensitivity, and resolution of the FP assay, which offers a more practical approach for identifying the hits and leads of PPI inhibitors.

In addition to the APC–Asef system, we extended the sensitivity-enhanced tracer strategy to the p53–MDM2 system. The original tracer in the FP assay for the p53–MDM2 system was PS-P53-11K with a $K_d$ value of 0.18 μM[15]. Based on the structure of p53–MDM2 complex (PDB: 1YCR), we hypothesized that a longer linker between FITC and the p53 peptide might introduce a binding site for the FITC group. Therefore, we designed and synthesized 4 tracers (PS-P53-7G, PS-P53-8H, PS-P53-9I, PS-P53-10J) with different lengths of the linker between FITC and the p53 peptide (Supplementary Table 6). We obtained the binding affinity ($K_d$) values of the 4 tracers by FP assays (Supplementary Table 6, Supplementary Fig. 18). Compared to the original tracer PS-P53-11K, PS-P53-9I exhibited an increased binding affinity ($K_d$ = 0.10 ± 0.008 μM) by 1.8-fold. To elucidate the binding mechanism of PS-P53-9I, we predicted the binding mode of PS-P53-9I and MDM2 using MD simulations. As expected, the FITC group of PS-P53-9I could adapt to the binding site at the back of MDM2 (Supplementary Fig. 19). The hydroxyl moiety of FITC in PS-P53-9I formed a hydrogen bond with the S66 and V51 of MDM2 separately. The oxygen atom of the xanthene moiety formed a hydrogen bond with the F67 of MDM2. The xanthene moiety of FITC established strong van der Waals interactions (<4.5 Å) with the Y52 and K74 residues. These interactions stabilized the FITC group and further improved the binding affinity of PS-P53-9I to MDM2. Then, we selected PS-P53-9I as a potent tracer to develop the FP assay and measured the IC$_{50}$ values of the inhibitors P53 (IC$_{50}$ = 4.3 ± 1.5 μM) and PS5 (IC$_{50}$ = 6.3 ± 1.5 μM) with similar activity measured by PS-P53-11K. In comparison with PS-P53-11K, PS-P53-9I clearly distinguished between inhibitors P53 and PS5 with IC$_{50}$ values of 4.8 ± 1.4 μM and 16 ± 1.2 μM, respectively (Supplementary Table 7, Supplementary Fig. 20). The successful design of PS-P53-11K also confirmed the effectiveness of the sensitivity-enhancement strategy.

## Methods

### Peptide synthesis and purification
Detailed synthetic procedures are available in Supplementary Methods.

### Plasmid construction
APC (303–739), APC (407–751) and Asef (170–632) were kind gifts from Dr. Geng Wu (Shanghai Jiao Tong University). The FLAG-tagged APC (303–876) and FLAG-tagged APC (1–1338) constructs were created by direct insertion of the APC gene in the pBABE-puro retroviral vector between the BamHI and SalI restriction sites. The HA-tagged Asef (170–632) construct was created by direct insertion of the Asef gene in the pBABE-puro retroviral vector between the BamHI and SalI restriction sites[16]. HA-tagged APC (303–876) and FLAG-tagged Asef (170–632) constructs were created by direct insertion of the APC or Asef gene in

the pcDNA 3.1 (+) vector between the XhoII and NotI restriction sites. Site-directed mutations were generated by Mut Express II fast mutagenesis kit (Vazyme; C214).

## Protein expression and purification

The DNA encoding APC (303–739), APC (407–751), and site-directed mutants (W593A, N641A) were cloned into pET28a expression vector and expressed as His-tagged fusion proteins in *Escherichia coli* (*E. coli*) strain BL21 (DE3)[16]. The primers were provided in Supplementary Table 8. The cells were harvested and resuspended in lysis buffer (25 mM Tris·HCl, pH 8.0, 300 mM NaCl, and 20 mM imidazole). After sonication, the cell lysates were centrifuged and purified by Ni-NTA affinity chromatography (GE Healthcare). The purity of APC (residues 303–739) was confirmed by sodium dodecyl sulfate–polyacrylamide gel electrophoresis (SDS-PAGE) analysis.

## FP-based direct binding and competitive assays

FITC-labeled tracer peptides were obtained from Changzhou Kanglong Biotech Ltd. (Changzhou, Jiangsu, China). To determine the binding of tracers to APC (303–739), the tracers (fixed at 20 nM) were incubated with a range of concentrations of APC protein. The FP signal changes upon tracer binding were measured after 1.5 h of incubation using a Synergy neo microplate reader (BioTek Instruments, Inc.) and plotted against the protein concentration. The corresponding $EC_{50}$ values were generated by fitting the curves with a nonlinear regression model. For the competitive assays, the final APC (303–739) concentration was set as 25 nM based on the $EC_{50}$ and $EC_{80}$ values determined from the direct binding assay. To make a working solution, APC (303–739) was diluted in FP buffer (50 mM HEPES, pH 7.5, 300 mM NaCl, 1 mM EDTA, 1 mM DTT). The diluted APC protein (91 μL) was combined with a serial dilution of peptides (4 μL) and incubated at room temperature for 1 h, followed by the addition of 5 μL of 400 nM tracer 7. The FP signals were measured after 1.5 h of incubation at room temperature using a Synergy neo microplate reader. Three controls were included on each plate, i.e., blank (without protein and tracer), 100% inhibition (free tracer only), and 0% inhibition (protein/tracer complex only) controls. All experiments were performed in triplicate. The $IC_{50}$ values of peptides were calculated by nonlinear regression analysis using GraphPad Prism 7.0 software[17].

## Surface plasmon resonance

SPR experiments were performed using a Biacore 8 K system equipped with a CM5 sensor chip. His-tagged APC (303–739) was immobilized using amine-coupling chemistry. The surfaces of the two flow cells were activated for 420 s with a 1:1 mixture of 0.1 M NHS and 0.1 M EDC at a flow rate of 5 μL min⁻¹. The ligand at a 10 μg mL⁻¹ concentration in 10 mM sodium acetate, pH 5.0, was immobilized on flow cell 2; flow cell 1 was left blank to serve as a reference surface. Both surfaces were blocked with a 420 s injection of 1 M ethanolamine, pH 8.0. To collect kinetic binding data, a serial concentration of peptide inhibitors in 1× PBS, pH 7.4, was injected over the two flow cells at a flow rate of 30 μL min⁻¹ and a temperature of 25 °C. The complex was allowed to associate and dissociate for 60–240 and 150–600 s, respectively. The experiments were performed in triplicate. The corresponding sensorgrams were corrected for the DMSO bulk response by using calibration curves obtained with a 0.6–1.4% DMSO running buffer. The data were fitted to a Langmuir 1:1 binding model using the local $R_{max}$ analysis option available in the Biacore insight evaluation software.

## Crystallization and structure determination

APC protein (407–751, 15 mg mL⁻¹) was used in the crystallography experiments. Crystals of the MAI-516–APC and tracer 7–APC complexes were obtained through the sitting-drop vapor-diffusion method by mixing 0.5 μL of the protein complex solution with 0.5 μL of the reservoir buffer and equilibrating the mixture against 70 μL of the

reservoir solution at 18 °C. The crystals were cryoprotected in 0.2 M ammonium sulfate, 0.1 M Tris pH 8.0, 25% (wt/vol) PEG 3350, and 15% glycerol and then flash cooled in liquid nitrogen. Diffraction data were collected at beamline BL17U1 of the Shanghai Synchrotron Radiation Facility. The structures were solved, further refined, and deposited into the PDB with codes 7F6M and 7F7O.

## ITC assay

ITC experiments were performed using a MicroCal ITC200 calorimeter (Malvern Panalytical Ltd.). APC protein and peptide inhibitors were diluted with buffer containing 50 mM HEPES pH 7.5, 300 mM NaCl and 1 mM EDTA. The cells were loaded with APC proteins at 10–20 μM. All injections were performed using an initial injection of 0.2 μL followed by 19 injections of 2 μL at 100–200 μM. All measurements were carried out at 30 °C. The data were analyzed using a single-site binding model within MicroCal ITC200 analysis software to generate the stoichiometry ($N$), enthalpy changes ($\Delta H$), entropy changes ($\Delta S$), and the association constant ($K$) of binding. The changes in free energy ($\Delta G$) were calculated ($\Delta G = \Delta H - T\Delta S$).

## Cell lines

The HEK293T, HIEC-6, SW480, SW620, RKO, DLD-1, HT-29, LoVo, Caco-2 and LS-174T cell lines were originally purchased from the American Type Culture Collection (ATCC). The LS-513, LS-1034, NCI-H508, NCI-H716 and SNU-C2B cell lines were kindly provided by Prof. Meiyu Geng from the Shanghai Institute of Materia Medica and Prof. Lei Chen from the Eastern Hepatobiliary Surgery Institute. All cell lines were validated by short tandem repeat (STR) profiling. The HEK293T, SW480, SW620, RKO, DLD-1, HT-29, LoVo, Caco-2 and LS-174T cells were cultured in Dulbecco's Modified Eagle Medium (DMEM)/high-glucose (HyClone) supplemented with 10% fetal bovine serum (FBS), 2 mM glutamine and 1% penicillin-streptomycin (Gibco) at 37 °C in 5% $CO_2$. The HIEC-6, LS-513, LS-1034, NCI-H508, NCI-H716, and SNU-C2B cells were cultured in RPMI1640 medium, and the LoVo cells were cultured in F12K medium. Except for the culture medium, the culture conditions were the same for the SW480 cells as for these cell lines. The SW480 cells stably expressing wild-type FLAG-APC 1–1338, FLAG-APC 1–1338 W593A or FLAG-APC 1–1338 N641A mutant were generated by lentiviral infection. Briefly, lentiviruses were produced by transfecting HEK293T cells with pBABE-puro FLAG-APC 1–1338 WT, W593A or N641A mutant plasmids and the helper plasmids psPAX2 and pMD2.G. Virus supernatant was collected twice at 30 and 54 h after transfection. Cells were infected with the virus supernatant in the presence of 8 μg/ml polybrene for 24 h. Then stable cells were selected by treatment with 2 μg/ml puromycin for one week before subsequential assays.

## Western blot and coimmunoprecipitation

For direct western blot analysis, cells were harvested in NP-40 lysis buffer (1% NP-40, 50 mM Tris-HCl at pH 7.5, 150 mM NaCl, 1 mM PMSF, 25 mM NaF, 1 mM $Na_3VO_4$) supplemented with cOmplete™ Protease Inhibitor Cocktail (Roche). For the exogenous Co-IP of APC–Asef in drug-treated cell lysate, HEK293T cells were seeded in six-well plates on Day 1 in fresh DMEM containing 10% FBS and transfected with pBABE puro FLAG-APC (303–876) and pBABE puro HA-Asef (170–632) by Neofect DNA transfection reagent on Day 2. Forty-eight hours after transfection, the cells were washed twice with cold PBS and lysed in NP-40 lysis buffer for 1 h on ice. The cell lysates were clarified by centrifugation at 20,000 × *g* for 10 min at 4 °C. The supernatant was incubated with DMSO, MAI-203 (25 μM), MAI-516 (0.2 μM), MAI-516 (1 μM) or MAI-516 (5 μM) for 2 h at 4 °C. Then, 30 μL of anti-FLAG M2 Affinity gel was added and incubated overnight at 4 °C using a vertical rotator. The next day, the samples were centrifuged (7,000 × *g* for 30 s at 4 °C), and the supernatant was discarded. The beads were washed four times with NP-40 lysis buffer, and the bound proteins were eluted

in SDS sample buffer for immunoblotting analysis and detected with anti-FLAG (Sigma; F1804, 1:1,000), anti-HA antibody (Sigma; H9658, 1:20,000) and goat anti-mouse IgG secondary antibody (Signalway antibody; L3032, 1:50,000). For the exogenous Co-IP of APC–Asef in drug-treated HEK293T cells, HEK293T cells were transfected with pcDNA-HA-APC (303–876) and pcDNA-FLAG-Asef (170–632). Twenty-four hours after transfection, peptides and peptidomimetics were added to the culture medium for 24 h. Then, the cells were harvested in 0.3% NP-40 lysis buffer, and the cell lysate was incubated with anti-FLAG® M2 affinity gel (Sigma; A2220) for 3 h at 4 °C. For endogenous immunoprecipitation, SW480, DLD1, and RKO cells were seeded in 10-cm cell culture dishes. At 50–60% confluence, the cells were incubated with different concentrations of MAIT-516 or MAIT-203 for 48 h before cell lysis. Then, the resulting cell lysate was incubated with APC antibody (Merck Millipore; MAB3785, 1:100) and control IgG overnight and protein A/G agarose for another 3 h at 4 °C. The precipitated proteins were eluted with 60 μL of 1× loading buffer after boiling for 10 min at 95 °C and detected using APC (Merck Millipore; MAB3785, 1:500) and Asef (Proteintech; 55213-1-AP, 1:1,000) antibodies for western blot analysis.

## Migration assay with RTCA-DP

Real-time cell migration was evaluated using the xCELLigence RTCA-DP system (Roche). Briefly, SW480 cells were serum-starved for 8 h, digested, and resuspended in serum-free medium. The wells in the upper chamber of CIM plate 16 were preequilibrated with 30 μL serum-free medium for 1 h and filled with 100 μL of $4 \times 10^4$ serum-starved SW480 cells per well. The wells in the lower chamber of CIM plate 16 were filled with complete medium. Peptides or peptidomimetics were added to the wells in the upper and lower chambers. Cell index values were detected every 15 min throughout the procedure, and the data were collected and analyzed with RTCA software (version 2.3).

## Migration assay through wound healing

Cells were seeded in 12-well plates at a density of $4 \times 10^5$ cells per well and grown until 90% confluence in DMEM supplemented with 10% FBS. The monolayers were scratched with a sterile pipette tip (QSP; T104TLS-Q). The cells were washed with 1× PBS and then cultured in culture medium containing 1% FBS and peptides or peptidomimetics. The corresponding photos were captured at the time of scratching and after 24–72 h.

## Transwell migration assay

Cells were serum-starved for 8 h and then digested and resuspended in serum-free medium. The migration assays were performed using Transwell chambers in 24-well plates (Corning #3422). Cells were seeded in the upper chambers at a density of $8 \times 10^4$ (SW480, RKO, SNU-C2B) or $1.5 \times 10^5$ cells (HIEC-6, LS-1034, Caco-2, DLD-1, HT-29, LoVo) per well in 200 μL of serum-free medium. The lower chambers were filled with 600 μL of complete medium. Peptides or peptidomimetics were added to both the upper and lower chambers. After 48 h, the Transwell inserts were washed twice with 1× PBS. The cells in the upper chambers of the Transwell inserts were gently removed using cotton-tipped applicators. The cells on the lower surface of the membrane were fixed with methanol and then stained with crystal violet for 10 min. The Transwell inserts were washed with distilled water to remove the excess crystal violet and then dried. The stained cells from different areas were imaged using light microscopy and counted using ImageJ (NIH).

## Cell cytotoxicity assay

A total of 1000 Cells were plated in a 96-well plate and treated with DMSO, MAIT-203 (50 μM) or MAIT-516 (50 μM) for 48 h. The cell viability was measured by the CCK-8 assay (Yeasen Biotechnology, Shanghai) according to the manufacturer's instructions.

## Cell apoptosis assay

SW480 and HIEC-6 cells were cultured in 6-well plates to 70–80% confluence. The cells were treated with DMSO or MAIT-516 (25 μM) for 48 h. Then, the cells were collected by trypsinization and washed with ice-cold phosphate buffered saline (PBS). The PI/Annexin V-FITC assay was performed using a Pharmingen™ Annexin V Apoptosis Detection Kit (BD Biosciences) according to the manufacturer's instructions. Briefly, the harvested cells were incubated in 300 μL of 1× binding buffer containing 5 μL Annexin V and 5 μL PI for 30 min at room temperature in the dark. The cell apoptosis was measured and analyzed by flow cytometry.

## Three-dimensional (3D) organoid culture and CellTiter-Glo 3D viability assay

All animal experiment protocols followed the laboratory animal guidelines and were approved by the Animal Experiment Ethics Committee of Xinhua Hospital Affiliated to Shanghai Jiao Tong University School of Medicine. Mouse organoid culture, patient-derived organoid (PDO) culture and organoid viability assays were performed according to the initial reference[43]. Mice were housed in pathogen-free and ventilated cages with free access to food and water ad libitum under a 12:12 h light–dark cycle at room temperature of 21 ± 2 °C and humidity between 45% and 65%. Whole intestines from four-week-old C57BL/6 mice were used to establish mouse organoids. Institutional review board approval and informed consent were obtained for all CRC samples used for establishing the CRC PDOs. Mouse organoids and PDOs were first established and cultured in 24-well plates and replated in 96-well plates for drug treatment. Twenty-four hours after replating, the culture medium was replaced with fresh medium containing DMSO or MAIT-516 (25 μM) for another 48 h of treatment. The organoid viability was measured by a CellTiter-Glo luminescent cell viability assay (Promega, USA). Relative viability was normalized to that in the DMSO group.

## Immunofluorescence staining assay (IFA)

SW480 cells were seeded on glass coverslips overnight and then treated with MAIT-516 (25 μM) for 24 h or 48 h. The coverslips were then washed twice with PBS, and the cells were fixed in 4% paraformaldehyde for 15 min. After permeabilization with 0.5% Triton X-100 at room temperature for 10 min, the cells were blocked with PBS containing BSA (5%) for 1 h and then incubated with β-catenin antibody (CST; D10A8, 1:1,000) for 2 h. After washing three times, FITC-conjugated secondary antibody (Invitrogen; A11008, 1:300) was applied for 1 h of incubation. After washing three times again, the nuclei were stained with DAPI for 30 min. Immunofluorescence was visualized by an Olympus IX81.

## RNA extraction and real-time PCR analysis

Total RNA was extracted using TRIzol reagent (Takara Bio Inc., Japan) following the manufacturer's instructions. Total RNA (1 μg) was reverse-transcribed into cDNA using the PrimeScript RT Kit (Takara Bio Inc., Japan). Quantitative real-time reverse transcription-PCR was performed using a SYBR Green kit (Vazyme, China). β-Actin mRNA was employed as a reference gene for expression normalization. The qPCR primer sequences are listed in Supplementary Table 8.

## Statistical analysis

Quantitative data are presented as the mean ± SD or standard error of the mean (s.e.m.), as specified in the Table and Figure legends. Statistical tests were performed using GraphPad Prism 7 (GraphPad) or IBM SPSS Statistics 24 (IBM). For the experiments with two groups, a two-tailed unpaired Student's t test was performed to assess statistical significance. For the experiments with more than two groups, one-way analysis of variance (ANOVA) with post hoc Tukey's multiple comparisons test was used for multiple comparisons. For the RTCA data,

two-way ANOVA with post hoc Tukey's multiple comparisons test was used for multiple comparisons. Figures were assembled and formatted in Adobe Illustrator CC 2018 (Adobe Inc.).

## Reporting summary

Further information on research design is available in the Nature Research Reporting Summary linked to this article.

## Data availability

The structural data that support the findings in this study have been deposited in the Protein Data Bank with the coordinate accession numbers 7F6M and 7F7O. The previously published structural data used in this study are available in the Protein Data Bank under accession code 3NMZ, 1YCR, and 5Z8H. Uncropped blots and gels as well as the initial and final coordinates of the MD simulations are provided in the Source Data file. All other data generated or analyzed in this study are included in this article and its Supplementary Information file. Source data are provided in this paper.

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

## Acknowledgements

This work was supported in part by the National Natural Science Foundation of China (81925034 (to J.Zhang), 82172916 (to C.Y.L.)), the Innovation Program of Shanghai Municipal Education Commission (2019-01-07-00-01-E00036) (to J.Zhang), the Science and Technology Commission of Shanghai Municipality (19ZR1476400) (to X.Y.), YCTSQN2021011 (to X.Y.), UM Distinguished Visiting Scholar (to X.Y.), the Starry Night Science Fund of Zhejiang University Shanghai Institute for Advanced Study (SN-ZJU-SIAS-007) (to J.Zhang), innovative research team of high-level local universities in Shanghai (SSMU-ZLCX20180702 and SHSMU-ZDCX20212700) (to J.Zhang), the open fund of state key laboratory of Pharmaceutical Biotechnology, Nanjing University (KF-202204) (to J.Zhang), CAMS Innovation Fund for Medical Sciences (CIFMS) (2019-I2M-5-051) (to G.C.), GuangCi Professorship Program of Ruijin Hospital Shanghai Jiao Tong University School of Medicine (to J.Zhang), the Key Research and Development Program of Ningxia Hui Autonomous Region (2022CMG01002) (to J.Zhang).

## Author contributions

J.Zhang, X.Y., and C.Y.L. conceived the project. J.Zhong, Y.G., L.F., J.W., and Q.Z. performed the biological experiments and analyzed the data. J.Zhong generated the key protein reagents. X.Y. performed the peptide synthesis, purification, and characterization experiments. Z.Z. and K.S. performed crystallography experiments. K.S. solved the crystal structures. S.L. and Y.W. performed the MD simulations. G.C. provided suggestion for project design and article writing. All authors contributed to specific parts of the manuscript, with J.Zhang and X.Y. assuming responsibility for the manuscript in its entirety. J.C., P.S., and Y.S. synthesized and evaluated the peptides of the p53–MDM2 complex system.

## Competing interests

The authors declare no competing interests.
