## [Peer Review File · Nature Communications]

REVIEWER COMMENTS

Reviewer #1 (Remarks to the Author):

In their study, Yang, Zhang and colleagues report the structure-based affinity optimization of an APC-targeting peptide. APC is an attractive and so far elusive therapeutic target, and progress towards its druggability is certainly of general interest (due to its involvement in the formation of colorectal cancers). In preceding publications, the authors have reported peptide ligands that served as the starting point for this work. Here, they describe elegant and efficient ligand optimization efforts which have been verified with very strong biophysical data (crystal structures, SPR, and ITC). However, I see two major points that would have to be addressed before publication:

1) The authors perform an initial sequence optimization of the tracer peptide resulting in 13-fold increased affinity. A subsequent peptide linker truncation then provides another 2.5-fold increase in affinity. The authors apply the general workflow of optimizing a peptide ligand including some modelling efforts. To term this “double-anchor” strategy is overselling and should be removed.

2) The main relevance of this study arises from the potency of the reported new MAI derivatives. While their binding to APC has been characterized thoroughly. The cell-based characterization does not clearly verify an improved activity. Here, the three main issues:

- It is not explained why MAIT-203 was used and not the optimized inhibitor MAIT-400 (J. Med. Chem. 2018, 61, 8017–8028).

- It is not clear if the improved effect of MAIT-516 on cell migration (Figure 4c-e) is specific. Here, at least general cytotoxicity and the effect on wt APC cell lines should be tested.

- Why does Figure 4f not include a control MAIT derivative?

In addition, the following minor point should be addressed:

a) The introduction focuses on the challenges associated with the development of biochemical assays for the screening and the identification of PPI inhibitors and then stresses the importance of fluorescence polarization (FP) assays. This overstates the problem since i) There are highly sensitive readout systems to detect PPIs that are frequently used for screening purposes (e.g. HTRF and AlphaScreen), and ii) FP is not a common readout (at least not for high-throughput screening).

b) Along similar lines, the authors write: “Therefore, there is an urgent need for a robust tracer with higher sensitivity and a wider measurement range of inhibitor potency.” This is an exaggeration as competitive binding assays are usually used for primary screening, and hit validation is then performed with secondary assays which allow K_d determination and thereby affinity determination over a wide range (e.g. SPR or ITC).

c) The introduction should give more details regarding the importance of truncated APC both with respect to beta-catenin degradation and Asef activation, to inform the reader about the signalling context in particular with respect to colorectal cancer.

d) For Figure S1, it is not clear how that particular structure was generated.

e) For Figure 3b, the non-processed FP graphs should be added to the SI.

Reviewer #2 (Remarks to the Author):

The work by Zhang and co-authors provides an innovative solution to the long-standing problem of designing top-sensitivity tracers for evaluating protein-protein interactions with utility in both basic research and applied drug discovery and development. The double-anchor approach described within should find fairly broad applications in this large field because in my experience way too many drug discovery efforts targeting protein-protein interactions fail to reach a successful end point because the corresponding bioactivity assay uses a weak-affinity tracer. Here, the authors employ a clever strategy of utilizing the tracer's fluorophore as another binding partner that "touches" the protein target at a site adjacent to the binding pocket, which in turn provides a large increase in the favorable entropic component of the interaction to drive a dramatic improvement in the equilibrium binding constant. The new tracer that binds more tightly can in turn allow for a much more sensitive bioactivity assay.

The study appears to be very well executed and the manuscript is well written. The structural biology part of the work is crucial in proving that the double-anchor tracer works for the right reason, and its execution here strengthens the study significantly.

I recommend publication after a minor revision to address the following issue. The authors need to examine the paper for instances where numerical values are reported with too many or too few significant figures and synchronize their reporting accordingly. By way of example, on Line 140, a K_d value of 77.52 nM is reported. In the context of the methodology used to derive such a constant, the output of having 4 significant figures is unrealistic and could mislead many readers. A more appropriate reporting would be to list the K_d as 78 nM, that is to have 2 significant figures. In turn, in some places K_d or IC₅₀ values are listed with just one significant figure (eg, Line 157).

Reviewer #3 (Remarks to the Author):

In this manuscript, Zhong et al. are reporting a best-in-class peptide-based inhibitor of APC-Asef protein-protein interaction. Using an approach that the authors called “double-anchor” strategy, they optimized an existing fluorescence polarization (FP) tracer, which was used in SAR-based screening assays to identify novel peptide inhibitors with enhanced affinity for APC. As outlined by the authors, the APC-Asef interaction is critical for cell migration in colorectal cancer, which promotes tumor progression and metastasis.

The authors clearly identify the need for novel, more robust and sensitive high-affinity FP tracers as key problem in the field of protein-protein interaction inhibitor drug discovery. Through the design of their double-anchor strategy, the authors considered the possibility that the fluorophore group in the tracer could also participate to the target protein recognition and improve sensitivity of competitive-binding FP assays. This paper unfolds as two chapters. The first one is based on the description and optimization of tracer 7, as a tool to improve peptide screening for inhibitors of APC-Asef protein-protein interaction. It uses state-of-the-art techniques to evaluate ligand-target interactions, such as Surface Plasmon Resonance and Isothermal Titration Calorimetry. Beside optimizing the proximity of the fluorophore moiety on tracer 7 to increase the strength of the interaction with APC, this portion appears to me as an increment of previous articles published by this group. The second section of the paper focuses on the effects of a hit peptide (MAI-516) identified via FP-based screening assays using tracer 7 on colorectal cancer cell migration. This section is interesting but also quite preliminary and would requires further investigation. I have raised key questions and concerns that, hopefully, can be helpful to the authors to improve their manuscript.

Major points:

1) The authors must improve the storytelling of the paper, especially in the first portion of the manuscript. Unless one is well-aware of previous work from this group, it is very confusing and hard to understand which peptides were optimized based on what previous iterations. While this journal is targeting a broad readership, the authors are providing readers with very little context (e.g. Tracer 1, MAI-005...) to describe tracer optimization for APC-Asef interaction. I had to go read previous papers from this group to understand the origin of those “MAI” peptides and tracers used in FP assays.

2) In line 118, the authors mentioned “we shortened the linker between FITC and the peptide to increase the tracer rigidity and introduce a new binding site for the FITC group”. I understand that the NELISDG residues were removed from tracer 6 to generate tracer 7, but what is exactly meant by “introducing a new binding site for FITC”? Were there any modifications made to the FITC per se, to improve its “affinity” for APC (as I can understand the double-anchor principle illustrated in Figure.

1A)? If so, what is the nature of these modifications? This sounds critical to me since it raises the question whether it is possible to systematically adapt fluorophore's binding properties to any target of interest without impacting fluorescence integrity? Or is this simply a non-generalizable stroke of luck.

3) In line with the previous comment and based on the title and multiple other statements across the manuscript, this study should "move the needle" in the field of FP-based protein-protein interaction drug discovery by improving tracers' efficacy through the application of the "double-anchor approach". Thus, the authors should demonstrate other example(s) for the success of this approach. The example that FP tracers used for the identification of p53-MDM2 interaction disruptors are posing challenges was brought up by the authors in the introduction. Could such tracers be improved by a double-anchor strategy? Experimentally demonstrating this, using alpha-Helix-Mimicking Sulfono-gamma-AA peptide Inhibitors previously reported by Sang et al., 2020 (PMID: 31971801) would support such a major claim made by the authors.

4) In line 137, the authors wrote "On comparison with tracer 1, tracer 7 clearly distinguished the inhibitors with IC50 values less than 1 μ M (Figure 2d)" and in Figure. 2D legend "log-transformed half-maximal inhibitory concentration of 31 APC-Asef interaction inhibitors was determined". What are those 31 inhibitors? I can count 17 in Table-3 (with NMR data in Suppl. Material section) but these are related to the SAR described later. Those 31 inhibitors are brought up a bit out of nowhere by the authors and additional explanations are required.

5) In Figure. 3C, the binding kinetics between MAI-516 and APC was evaluated by Surface Plasmon Resonance, and "The equilibrium dissociation constant value of MAI-516 ($K_D = 18.2 \pm 2.1$ nM) was derived from the ratio between the kinetic dissociation (K_{off}) and association (K_{on}) constants obtained by fitting data from 0.39 to 50 nM using the simple 1:1 Langmuir binding fit model". Although a sensogram (RU vs. time) is presented in Figure. 3C, I could not find any fitting plots for K_D calculation (RU vs. conc.) in the manuscript. It would be appropriate to include it. Moreover, a comparative SPR dose-response assay using at least another APC-binding peptide would be important to show. The fitting for this comparative peptide should be superimposed (in the same graph) to the one obtained for MAI-516 (0.39 to 50 nM), and values added to Table-5.

6) Although it is informative on a biochemical point of view, the transfections of tagged constructs of APC and Asef fragments in HEK293T are not sufficient to confirm the inhibitory effect of MAIT-516 on APC-Asef interactions in different colorectal cancer cell models. The authors should add co-IP experiments in actual CRC cells, assessing endogenous full-length protein interactions in response to MAIT-516.

7) Moreover, the authors state that “cell-penetrating peptides (TAT) could increase the permeability of our peptides; hence, we conjugated the C terminus of MAI-516 with the optimized “GGGGG” linker and TAT to obtain the cell-permeable inhibitor MAIT-516”. Based on the data presented in Figure. 4A and B, it is not clear whether the addition of the TAT group is improving APC-Asef complex disruption. I would expect MAI-516 to be less potent than MAIT-516 when used in cell treatments. Quantitative analysis of the co-IPs would help clarifying this. If there are no changes in the levels of APC-Asef interactions for MAI-516 vs. MAIT-516, then, MAI-516 should also be tested in migration assays. However, if TAT+linker addition significantly increases the potency of MAI-516 in cells, it would become unclear whether the addition of TAT and the linker functions has an impact on the affinity of MAIT-516 for APC binding. Then, SPR and/or ITC analyses comparing the interaction of MAI-516 vs. MAIT-516 to APC would be important to report.

8) Investigations on the relevance of the W593A and N641A mutants in the context of colorectal cancer cells are missing. The authors should transduce colorectal cancer cell lines with overexpression vectors including both mutants and test cell migration in response to MAIT-516.

9) Different colorectal cell lines are known to express different truncated forms of the APC protein. Although most changes in APC in colorectal cancer lines were reported in portions beyond residue 876 (i.e. not in the domain participating to interaction with Asef), the authors tested basic cell functions such as migration (Figure. 4) and proliferation (Suppl. Fig. S7) in only one colorectal cancer cell line tested (SW480) using MAIT-516. This is not sufficient to support the authors’ claim on cell migration. These experiments should be expanded to other cell lines and, ideally, in primary patient tissues.

10) I understand the b-catenin-independent aspect of APC-Asef interactions in colorectal cancer, however, to the best of my knowledge however, it is not clear whether decreasing APC-Asef interactions in colorectal cancer cells will show no impact on the amounts/frequency of other protein-protein interactions influencing the canonical Wnt signaling pathway. The authors should at least test potential changes in nuclear/cytoplasmic b-catenin levels in response to MAIT-516, its level of ubiquitination/degradation, or changes in transcriptional activity (e.g. TOP flash luciferase assays). Moreover, the impact of their novel peptide on other critical functions, such as apoptosis, differentiation and tumorigenesis should also be assessed.

11) I understand that the authors are considering HEK293T cells as a normal cell model (see Figure S7 title). In many contexts in cell biology and drug discovery, HEK293T should not be used as a normal cell line, and I consider critical for the authors to assess the impact of MAIT-516 in a bona fide normal colon cell model vs. cancer cell lines, for migration and proliferation/growth assays. This could be done using primary colonic epithelial cells or established models described in the literature. This would be critical to confirm the cancer-specific relevance of MAIT-516 in the development of future therapeutic approaches.

Minor Points:

1) The paper needs a thorough proofreading and English editing job.

2) It looks odd to me to have data description with figure callouts in the introduction

3) In line 292 (Figure S7 legend) "CCK8 assays on cell proliferation after the treatment of MAI-516 in colorectal cancer cells and normal cells": It should say "MAIT-516" instead of MAI-516. There's also a typo in "normal".

Response to Reviewer #1's comments:

Summary: In their study, Yang, Zhang and colleagues report the structure-based affinity optimization of an APC-targeting peptide. APC is an attractive and so far elusive therapeutic target, and progress towards its druggability is certainly of general interest (due to its involvement in the formation of colorectal cancers). In preceding publications, the authors have reported peptide ligands that served as the starting point for this work. Here, they describe elegant and efficient ligand optimization efforts which have been verified with very strong biophysical data (crystal structures, SPR, and ITC). However, I see two major points that would have to be addressed before publication:

Response: Thanks for your kind comments and good suggestions. Firstly, to better describe the study for readers, we have removed inappropriate terms and sentences for the tracer strategy of FP and provided more details regarding the context of the truncated APC–Asef system in particular with respect to colorectal cancer (CRC) in the Introduction and Discussion sections of the revised manuscript. Secondly, we have assessed the cytotoxicity of MAIT-516 in cells and found that MAIT-516 at concentrations up to 50 μ M had no effect on cell growth of normal intestinal epithelial cells and colorectal cancer cells expressed full-length or truncated APC. Moreover, we examined the specific activity of MAIT-516 on cell migration by wound healing, transwell migration, and RTCA assays. The results showed that MAIT-516 at 25 μ M significantly attenuated the migration of additional colorectal cancer cells (Caco-2, DLD-1, HT-29, LoVo) expressed truncated APC, but had no apparent effect on normal intestinal epithelial cells (IEC-6) and colorectal cancer cells (SNU-C2B, RKO, LS-1034) with wild type (wt) APC, supporting the specific and improved activity of MAIT-516 on the truncated APC for cell migration.

Comment 1. The authors perform an initial sequence optimization of the tracer peptide resulting in 13-fold increased affinity. A subsequent peptide linker truncation then provides another 2.5-fold increase in affinity. The authors apply the general workflow of optimizing a peptide ligand including some modelling efforts. To term this “double-anchor” strategy is overselling and should be removed.

Response: We agree with your suggestion. We have removed the “double-anchor” statement in the manuscript.

Comment 2. The main relevance of this study arises from the potency of the reported new MAI derivatives. While their binding to APC has been characterized thoroughly. The cell-based characterization does not clearly verify an improved activity. Here, the three main issues:

a) It is not explained why MAIT-203 was used and not the optimized inhibitor MAIT-400 (*J. Med. Chem.* 2018, 61, 8017-8028).

Response: In our previous studies, MAI-203 was developed to block APC–Asef interaction in vitro, and conjugated TAT to generate the cell-penetrated MAIT-203, which is able to disrupt APC–Asef interaction in cell-based characterization (*Nat Chem Biol*, 2017, 13: 994-1001). Then, we reported an optimized inhibitor MAI-400 with better binding affinity to APC in vitro and cell lysates. However, perhaps due to the high polar surface area, we have not been able to synthesize a high-purity and cell-penetrated MAIT-400 through the conjugation of TAT to MAI-400, resulting in no such a MAIT-400 in publication (*J Med Chem*, 2018, 61: 8017-8028). Therefore, we used MAIT-203 as a control in cell-based assays of this study.

b) It is not clear if the improved effect of MAIT-516 on cell migration (Figure 4c-e) is specific. Here, at least general cytotoxicity and the effect on wt APC cell lines should be tested.

Response: Thanks, it is a good suggestion. To investigate the specific effects on colorectal cancer cells, we firstly evaluated the general cytotoxicity of MAIT-516 in normal intestinal epithelial cells (IEC-6) and colorectal cancer (CRC) cells expressed full-length (RKO, LS-513, LS-1034, and SNU-C2B) or truncated (Caco-2, HT-29, LoVo, DLD-1, SW480, and SW620) APC using the CCK-8 assay as previously described (*Nat Chem Biol*, 2014, 10: 298-304; *Nat Commun*, 2014, 5: 3067). Our result showed that both MAIT-203 and MAIT-516 at concentrations up to 50 μ M did not impair the viability of normal intestinal epithelial cells and CRC cells, regardless of which type of APC was expressed (Supplementary Fig. 15a) (page 13, lines 293-298). Furthermore, to investigate the specificity of the inhibitory effect of MAIT-516 on CRC cell migration, we performed wound healing and transwell migration assays with normal intestinal epithelial cells and CRC cells expressing full-length or truncated APC. Our results showed that MAIT-516 had no noticeable influence on the migration of either normal intestinal epithelial cells (hIEC-6) (Supplementary Fig. 10) or CRC cells expressing full-length APC (LS-1034, RKO, SNU-C2B) (Supplementary Fig. 11). In contrast, MAIT-516 significantly restrained the migration of CRC cells expressing truncated APC (Caco-2, DLD-1, HT-29, LoVo) at 25 μ M (Supplementary Fig. 12) in addition to reducing the migration of SW480 cells. These data indicated that the suppressive effect of MAIT-516 on cell migration was specific to CRC cells with truncated APC expression (page 12, lines 264-272).

c) Why does Figure 4f not include a control MAIT derivative?

Response: We have added the MAIT-203 as a control in the xCELLigence Real-Time Cell Analysis (RTCA) assay as described in Fig. 4e. The results showed that the anti-migration effect of MAIT-516 at 10 μ M was more potent than that

of MAIT-203, which was in agreement with the quantitative binding affinities of the inhibitors to APC (page 11, lines 256-257).

Collectively, the above results elucidated that MAIT-516 with improved activity compared to MAIT-203 specifically inhibited the metastasis of CRC cells expressing truncated APC by the suppression of APC–Asef-mediated cell migration (page 14, lines 313-315). We have added the result and discussion in the revised manuscript (page 11, lines 256-257; page 12, lines 264-272; page 13, lines 293-298; page 14, lines 313-315).

Comment 3. The introduction focuses on the challenges associated with the development of biochemical assays for the screening and the identification of PPI inhibitors and then stresses the importance of fluorescence polarization (FP) assays. This overstates the problem since i) There are highly sensitive readout systems to detect PPIs that are frequently used for screening purposes (e.g. HTRF and AlphaScreen), and ii) FP is not a common readout (at least not for high-throughput screening).

Response: Thanks for your kind indication. To correct the previous overstatement, we have revised the introduction to focus on the improvement of FP assay in PPI inhibitor screening instead of highlighting the importance of FP (page 3, lines 56-59; page 4, lines 60-71). Indeed, biochemical systems for the identification of PPI inhibitors include fluorescence polarization (FP) assay, fluorescent resonance energy transfer (FRET) assay, AlphaScreen, and enzyme linked immunosorbent (ELISA), and each of these methods has its own advantages and disadvantages in a variety of PPI targets. According to your suggestion, we have discussed the characteristics of these readout systems in the Discussion section of the revised manuscript (page 14, lines 328-329; page 15, lines 330-343).

Comment 4. Along similar lines, the authors write: “Therefore, there is an urgent need for a robust tracer with higher sensitivity and a wider measurement range of inhibitor potency.” This is an exaggeration as competitive binding assays are usually used for primary screening, and hit validation is then performed with secondary assays which allow K_d determination and thereby affinity determination over a wide range (e.g. SPR or ITC).

Response: We agree with the comment of the reviewer. We exactly used SPR and ITC as secondary assays to validate the activity of our inhibitor MAI-516 in the study. We have removed the inappropriate statement in the revised manuscript.

Comment 5. The introduction should give more details regarding the importance of truncated APC both with respect to beta-catenin degradation and Asef activation, to inform the reader about the signalling context in particular with respect to colorectal cancer.

Response: Thank you for the suggestion. APC is responsible for multiple signaling pathways via various protein interactions. Commonly, APC promotes the degradation of β -catenin and thereby negatively regulates Wnt signaling. In colorectal cancer (CRC) metastasis, APC gene alterations were observed to generate C-terminal truncated APC proteins, which caused the loss of the domains required for β -catenin binding but increased Asef activation through the APC-ARM domain. The Asef guanine nucleotide exchange factor (GEF) activity constitutively activated by APC-ARM binding promoted CRC cell migration via small Rho-like GTPase signaling. We added the signaling context on APC both with respect to beta-catenin degradation and Asef activation in the Introduction section of the revised manuscript (page 3, lines 47-53).

Comment 6. For Figure S1, it is not clear how that particular structure was generated.

Response: Thank you for the indication. The particular binding structures of tracers 1, 4, and 5 in Figure S1 are generated by molecular modeling and molecular dynamics (MD) simulations as follows: (1) constructed three initial tracer-APC complexes by mutating residues and adding FITC motif into the “GGGGEQLAINELISDGS” segment from Asef based on the crystal structure (PDB ID: 3NMZ) resolved in our previous work (*Cell Res*, **2012**, 22: 372-386), followed by energy minimization of the systems; (2) created the topology files and coordinate files of the three systems for the simulations, then built the water box with a cube with sides of 10 Å for each system and added counter ions (Na^+ or Cl^-) to keep the systems in electric neutrality; (3) performed the MD simulations based on the workflow in our previous work (*Chem. Sci*, **2021**, 12: 464-476); (4) analyzed the trajectories from each simulation by Pymol and selected a representative conformation of the best energy cluster as the particular structure shown in Supplementary Fig. 1. We have added the procedure in the revised Supplementary Information (page 11, lines 333-340).

Comment 7. For Figure 3b, the non-processed FP graphs should be added to the SI.

Response: We have added the non-processed FP graphs of Fig. 3b as Supplementary Fig. 4 in the revised Supplementary Information (page 13, lines 376-377).

Response to Reviewer #2's comments:

Summary: The work by Zhang and co-authors provides an innovative solution to the long-standing problem of designing top-sensitivity tracers for evaluating protein-protein interactions with utility in both basic research and applied drug discovery and development. The double-anchor approach described within should find fairly broad applications in this large field because in my experience way too many drug discovery efforts targeting protein-protein interactions fail to reach a successful end point because the corresponding bioactivity assay uses a weak-affinity tracer. Here, the authors employ a clever strategy of utilizing the tracer's fluorophore as another binding partner that "touches" the protein target at a site adjacent to the binding pocket, which in turn provides a large increase in the favorable entropic component of the interaction to drive a dramatic improvement in the equilibrium binding constant. The new tracer that binds more tightly can in turn allow for a much more sensitive bioactivity assay. The study appears to be very well executed and the manuscript is well written. The structural biology part of the work is crucial in proving that the double-anchor tracer works for the right reason, and its execution here strengthens the study significantly. I recommend publication after a minor revision to address the following issue.

Response: Thank you for the insights provided in consideration of our manuscript. We have synchronized the numerical values with the same significant figures in the revised manuscript.

Comment 1. The authors need to examine the paper for instances where numerical values are reported with too many or too few significant figures and synchronize their reporting accordingly. By way of example, on Line 140, a K_d value of 77.52 nM is reported. In the context of the methodology used to derive such a constant, the output of having 4 significant figures is unrealistic and could mislead many readers. A more appropriate reporting would be to list the K_d as 78 nM, that is to have 2 significant figures. In turn, in some places K_d or IC_{50} values are listed with just one significant figure (eg, Line 157).

Response: Thanks for your kind suggestion. We have carefully checked the significant figures throughout the manuscript, and synchronized the reporting significant figures. In particular, we corrected a K_d or IC_{50} value with 2 significant values, consistent with the methodology used in the revised manuscript (page 2, lines 24 and 26; page 4, lines 68 and 76; page 5, lines 89 and 90; page 6, lines 114, 115, 118, 120, 125 and 127; page 8, lines 164,

170-172 and 178; page 9, lines 185-187 and 192; page 10, line 232; page 14, line 320; page 16, 363, 369, 377 and 379; Tables 1-6).

Response to Reviewer #3's comments:

Summary: In this manuscript, Zhong et al. are reporting a best-in-class peptide-based inhibitor of APC-Asef protein-protein interaction. Using an approach that the authors called “double-anchor” strategy, they optimized an existing fluorescence polarization (FP) tracer, which was used in SAR-based screening assays to identify novel peptide inhibitors with enhanced affinity for APC. As outlined by the authors, the APC-Asef interaction is critical for cell migration in colorectal cancer, which promotes tumor progression and metastasis. The authors clearly identify the need for novel, more robust and sensitive high-affinity FP tracers as key problem in the field of protein-protein interaction inhibitor drug discovery. Through the design of their double-anchor strategy, the authors considered the possibility that the fluorophore group in the tracer could also participate to the target protein recognition and improve sensitivity of competitive-binding FP assays. This paper unfolds as two chapters. The first one is based on the description and optimization of tracer 7, as a tool to improve peptide screening for inhibitors of APC-Asef protein-protein interaction. It uses state-of-the-art techniques to evaluate ligand-target interactions, such as Surface Plasmon Resonance and Isothermal Titration Calorimetry. Beside optimizing the proximity of the fluorophore moiety on tracer 7 to increase the strength of the interaction with APC, this portion appears to me as an increment of previous articles published by this group. The second section of the paper focuses on the effects of a hit peptide (MAI-516) identified via FP-based screening assays using tracer 7 on colorectal cancer cell migration. This section is interesting but also quite preliminary and would requires further investigation. I have raised key questions and concerns that, hopefully, can be helpful to the authors to improve their manuscript.

Response: Thank you for the insights provided in consideration of our manuscript. According to your suggestion, we evaluated the effect of MAIT-516 on APC-Asef interaction in colorectal cancer cells with wild type (wt)/truncated APC, and the result showed that MAIT-516 did not obviously affect the cell migration in cell lines with wt APC, compared to significant inhibition of cell migration in colorectal cancer cells expressing truncated APC. Then, we found that MAIT-516 did not suppress the cell proliferation of 15 colorectal epithelial cell lines with different genetic backgrounds, and showed no cytotoxicity in primary organoid models derived from mouse intestines and paired human normal and colorectal cancer tissues. Collectively, the results indicate that MAIT-516 specifically inhibited the metastasis of colorectal cancer cells expressed truncated APC by the suppression of APC-Asef mediated cell migration. In addition, using the strategy in the p53-MDM2 system, we also designed a tracer with

enhanced activity and improved sensitivity of competitive-binding FP assay for p53-MDM2 inhibitors, extending the application of the strategy.

Comment 1. The authors must improve the storytelling of the paper, especially in the first portion of the manuscript. Unless one is well-aware of previous work from this group, it is very confusing and hard to understand which peptides were optimized based on what previous iterations. While this journal is targeting a broad readership, the authors are providing readers with very little context (e.g. Tracer 1, MAI-005...) to describe tracer optimization for APC-Asef interaction. I had to go read previous papers from this group to understand the origin of those “MAI” peptides and tracers used in FP assays.

Response: Thank you for the indication. In our previous studies, we established an FP competitive assay to screen inhibitors of APC–Asef interaction. The FP assay used a FITC-labelled peptide as a tracer (Ac-¹⁷⁶GGGGEQQLAINELISDGK¹⁹⁵[FITC]–NH₂, Tracer 1). Using FP assay, we screened a series of truncated peptides based on the residues 176-194 of Asef and then found an inhibitor hit MAI-005 (¹⁸¹GGEQLAI¹⁸⁷) of APC–Asef interaction. Then, we optimized MAI-005 into two peptidomimetic inhibitors, MAI-203 and MAI-400. To aid in the translocation of peptidomimetic inhibitors across mammalian cell plasma membranes, we conjugated MAI-203 with a transcriptional transactivating (TAT) sequence and yielded MAIT-203, which inhibited CRC cell migration by disrupting the APC–Asef PPI. We have added this background to the Introduction of the revised manuscript (page 3, lines 56-59; page 4, lines 60-64).

Comment 2. In line 118, the authors mentioned “we shortened the linker between FITC and the peptide to increase the tracer rigidity and introduce a new binding site for the FITC group”. I understand that the NELISDG residues were removed from tracer 6 to generate tracer 7, but what is exactly meant by “introducing a new binding site for FITC”? Were there any modifications made to the FITC per se, to improve its “affinity” for APC (as I can understand the double-anchor principle illustrated in Figure. 1A)? If so, what is the nature of these modifications? This sounds critical to me since it raises the question whether it is possible to systematically adapt fluorophore’s binding properties to any target of interest without impacting fluorescence integrity? Or is this simply a non-generalizable stroke of luck.

Response: Thank you for the comment. Instead of modifying the structure of FITC itself, in the study, we optimized the length of the linker between FITC and the peptide (Supplementary Table 1, Supplementary Fig. 2), leading to the formation of additional interaction (e.g. hydrogen bond) between FITC and potential pocket of APC. According to previous reports, there are some pieces of evidence for the roles of FITC to its target. For example, Jan et al.

showed that the addition of FITC improved the binding affinities of their inhibitors to Glutamate carboxypeptidase II (*J Med Chem*, **2015**, 58: 4357-4363). Vincenzo et al. found the change of linker between FITC and inhibitor sulfonamide caused the different activity of inhibition against Carbonic anhydrase (*J Am Chem Soc*, **2006**, 128: 8329-8335). These findings indicate that as a moiety of tracer, FITC not only offers the fluorescence for real-time detection but could provide a new binding contribution of the tracer to the target, resulting in the improvement of the sensitivity of the FP assay. We have polished the ambiguous statement in the revised manuscript (page 5, lines 101-105; page 6, lines 123-124; page 15, lines 348-354; page 16, lines 355-360).

In addition to APC–Asef complex, we extended the sensitivity-enhanced tracer strategy in the p53-MDM2 system. Firstly, we optimized the original tracer PS-P53-11K in the p53-MDM2 system (*J Med Chem*, **2020**, 63: 975-986) based on the complex structure of p53-MDM2 (PDB: 1YCR). By the evaluation of new tracers with different lengths of the linker between FITC and the p53 peptide (Supplementary Table 6, Supplementary Fig. 19), we obtained an improved tracer PS-P53-9I with 1.8-fold increase of binding affinity to MDM2. Molecular modeling showed that PS-P53-9I could form additional interaction by the FITC binding to the area at the back of MDM2 (Supplementary Fig. 20). In comparison with the original tracer PS-P53-11K, tracer PS-P53-9I clearly distinguished the inhibitors P53 and PS5 with IC_{50} values of $4.8 \pm 1.4 \mu\text{M}$ and $16 \pm 1.2 \mu\text{M}$ respectively (Supplementary Table 7, Supplementary Fig. 21), indicating that the sensitivity-enhanced strategy works well in the p53-MDM2 system. We have added the results in the revised manuscript (page 16, lines 361-379; page 17, lines 380-381).

Comment 3. In line with the previous comment and based on the title and multiple other statements across the manuscript, this study should “move the needle” in the field of FP-based protein-protein interaction drug discovery by improving tracers’ efficacy through the application of the “double-anchor approach”. Thus, the authors should demonstrate other example(s) for the success of this approach. The example that FP tracers used for the identification of p53-MDM2 interaction disruptors are posing challenges was brought up by the authors in the introduction. Could such tracers be improved by a double-anchor strategy? Experimentally demonstrating this, using **alpha-Helix-Mimicking Sulfonyl-gamma-AA peptide Inhibitors previously reported by Sang et al., 2020 (PMID: 31971801)** would support such a major claim made by the authors.

Response: Thank you for the suggestion. As mentioned in *Comment 2*, we performed the sensitivity-enhanced strategy on alpha-Helix-Mimicking Sulfonyl-

gamma-AA peptide inhibitors in the p53-MDM2 system previously reported by Sang et al., 2020 (PMID: 31971801). The new tracer PS-P53-9I showed a 1.8-fold binding affinity to MDM2 than the original tracer, and the FP assay using the improved PS-P53-9I exhibited better sensitivity in the measure of p53-MDM2 inhibitors P53 and PS5 whose activities are indistinguishable by the original tracer. We have added the result and discussion in the revised manuscript (page 16, lines 361-379; page 17, lines 380-381).

Comment 4. In line 137, the authors wrote “On comparison with tracer 1, tracer 7 clearly distinguished the inhibitors with IC₅₀ values less than 1 μM (Figure 2d)” and in Figure. 2D legend “log-transformed half-maximal inhibitory concentration of 31 APC-Asef interaction inhibitors was determined”. What are those 31 inhibitors? I can count 17 in Table-3 (with NMR data in Suppl. Material section) but these are related to the SAR described later. Those 31 inhibitors are brought up a bit out of nowhere by the authors and additional explanations are required.

Response: Thank you for the indication. In Figure 2d, those 31 inhibitors consist of compounds in our previous and current manuscripts as well as some unpublished compounds in our patent (ZL201610330969.4, 202110017194.6). To make SAR more complete, we have added the activities of the 31 inhibitors measured by tracer 1 and 7 in Supplementary Table 4. Meanwhile, the structures and the spectra result of these compounds are provided in Supplementary Table 5 (pages 29-32, lines 501-505).

Comment 5. In Figure. 3C, the binding kinetics between MAI-516 and APC was evaluated by Surface Plasmon Resonance, and “The equilibrium dissociation constant value of MAI-516 ($K_D = 18.2 \pm 2.1$ nM) was derived from the ratio between the kinetic dissociation (K_{off}) and association (K_{on}) constants obtained by fitting data from 0.39 to 50 nM using the simple 1:1 Langmuir binding fit model”. Although a sensogram (RU vs. time) is presented in Figure. 3C, I could not find any fitting plots for K_D calculation (RU vs. conc.) in the manuscript. It would be appropriate to include it. Moreover, a comparative SPR dose-response assay using at least another APC-binding peptide would be important to show. The fitting for this comparative peptide should be superimposed (in the same graph) to the one obtained for MAI-516 (0.39 to 50 nM), and values added to Table-5.

Response: According to your suggestion, we have added the SPR sensorgram of the control peptide MAI-203 in Fig. 3c. Meanwhile, the affinity fitting plots of MAI-203 and MAI-516 are provided in Fig. 3e, and their kinetic data from SPR has been added to Table 5.

Comment 6. Although it is informative on a biochemical point of view, the transfections of tagged constructs of APC and Asef fragments in HEK293T are not sufficient to confirm the inhibitory effect of MAIT-516 on APC-Asef interactions in different colorectal cancer cell models. The authors should add co-IP experiments in actual CRC cells, assessing endogenous full-length protein interactions in response to MAIT-516.

Response: Following the reviewer's suggestions, we performed the endogenous Co-IP of APC-Asef in response to MAIT-516 in CRC cells expressing truncated APC (SW480, DLD1) and WT full-length APC (RKO). Consistent with the results of overexpressed tagged proteins in HEK293T cells, the interaction between endogenous truncated APC and Asef was dramatically diminished by MAIT-516 treatment in both SW480 and DLD1 cells (new Fig. 4b, Supplementary Fig. 8b). In contrast, MAIT-516 showed little effect on the endogenous PPI between WT APC and Asef (Supplementary Fig. 8c), which demonstrated the specific inhibition of the truncated APC-Asef PPI by MAIT-516. We have added the results in the revised manuscript (page 11, lines 243-249).

Comment 7. Moreover, the authors state that "cell-penetrating peptides (TAT) could increase the permeability of our peptides; hence, we conjugated the C terminus of MAI-516 with the optimized "GGGGG" linker and TAT to obtain the cell-permeable inhibitor MAIT-516". Based on the data presented in Figure. 4A and B, it is not clear whether the addition of the TAT group is improving APC-Asef complex disruption. I would expect MAI-516 to be less potent than MAIT-516 when used in cell treatments. A quantitative analysis of the co-IPs would help clarify this. If there are no changes in the levels of APC-Asef interactions for MAI-516 vs. MAIT-516, then, MAI-516 should also be tested in migration assays. However, if TAT+linker addition significantly increases the potency of MAI-516 in cells, it would become unclear whether the addition of TAT and the linker functions have an impact on the affinity of MAIT-516 for APC binding. Then, SPR and/or ITC analyses comparing the interaction of MAI-516 vs. MAIT-516 to APC would be important to report.

Response: Yes, we agree with your comments. To investigate the effect of TAT, we compared the treatment of MAI-516 and its TAT-conjugated derivative MAIT-516 in cells. Co-IP of exogenous HA-APC (303-876) and Flag-Asef (170-632) in HEK293T cells showed that MAI-516 and TAT+linker (50 μ M) had no effect on APC-Asef interaction, while MAIT-516 significantly inhibited APC-Asef interaction in a dose-dependent manner (Fig. 4a), indicating the improvement of TAT group in the permeability of cell membranes. In addition, we demonstrated that MAI-516 had no apparent effect on the migration of SW480 cells in wound-healing,

transwell, and RTCA migration assays (Supplementary Fig. 9), whereas MAIT-516 significantly inhibited the migration of SW480 cells (Fig. 4c-g), which was consistent with the results from Co-IP assays. To further evaluate the TAT+linker addition, we measured the binding affinity of MAI-516, TAT+linker, and MAIT-516 to APC using ITC assays. The analyses showed that there was no obvious binding between TAT+linker and APC, and the dissociation constant K_d of MAI-516 and MAIT-516 were 4.4 nM and 67 nM, respectively (Fig. 3f, Supplementary Fig. 7d, 7e, and Table 6), demonstrating that TAT+linker is solely responsible for cell penetration. We have added the results in the revised manuscript (page 10, lines 229-233; page 11, lines 234-240; page 12, lines 259-263).

Comment 8. Investigations on the relevance of the W593A and N641A mutants in the context of colorectal cancer cells are missing. The authors should transduce colorectal cancer cell lines with overexpression vectors including both mutants and test cell migration in response to MAIT-516.

Response: It is a good suggestion. To investigate the mutants in the context of colorectal cancer cells, we constructed SW480 stable cells overexpressing truncated APC (1-1338, the same as the endogenous APC expressed in SW480 cells) and the W593A and N641A mutants (Supplementary Fig. 13a). Along with previous reports (*Nat Cell Biol*, **2003**, 5: 211-215), the overexpression of truncated APC dramatically enhanced the migration of SW480 cells (Supplementary Fig. 13b, c). Moreover, the W593A mutant almost abrogated the enhanced cell migration induced by overexpression of truncated APC, while the N641A mutant diminished the promigratory activity of truncated APC to a lesser extent (Supplementary Fig. 13b, c). In addition, MAIT-516 significantly reduced the migration of SW480 cells overexpressing truncated WT APC but not that of cells overexpressing W593A or N641A mutants (Supplementary Fig. 13b, c). These data further supported the vital role of W593/N641 in increasing the binding affinity of MAI-516 to APC. We have added the results in the revised manuscript (page 12, lines 273-282; page 13, line 283).

Comment 9. Different colorectal cell lines are known to express different truncated forms of the APC protein. Although most changes in APC in colorectal cancer lines were reported in portions beyond residue 876 (i.e. not in the domain participating to interaction with Asef), the authors tested basic cell functions such as migration (Figure. 4) and proliferation (Suppl. Fig. S7) in only one colorectal cancer cell line tested (SW480) using MAIT-516. This is not sufficient to support the authors' claim on cell migration. These experiments should be expanded to other cell lines and, ideally, in primary patient tissues.

Response: According to your suggestions, we tested the effect of MAIT-516 on cell migration and proliferation in 4 additional CRC cell lines expressing truncated APC (Caco-2, DLD-1, HT-29, LoVo), 3 CRC cell lines with WT APC (RKO, LS-1034, SNU-C2B) and normal intestinal epithelial cells (hIEC-6). Our results showed that MAIT-516 had no noticeable influence on the migration of either normal intestinal epithelial cells (hIEC-6) (Supplementary Fig. 10) or CRC cells expressing full-length APC (LS-1034, RKO, SNU-C2B) (Supplementary Fig. 11). In contrast, MAIT-516 significantly restrained the migration of CRC cells expressing truncated APC (Caco-2, DLD-1, HT-29, LoVo) at 25 μ M (Supplementary Fig. 12) in addition to reducing the migration of SW480 cells. These data indicated that the suppressive effect of MAIT-516 on cell migration was specific to CRC cells with truncated APC expression. In addition, MAIT-516 did not suppress the cell proliferation of 15 colorectal epithelial cell lines with different genetic backgrounds (Supplementary Fig. 15a). To expand the evaluation on patient tissues, we further assessed the toxicity of MAIT-516 in the primary organoids derived from mouse intestines and paired human normal and colorectal tissues. Consistently, MAIT-516 showed little cytotoxicity in these primary organoid models (Supplementary Fig. 15b-f). We have added the results in the revised manuscript (page 12, lines 264-272; page 13, lines 293-301).

Comment 10. I understand the b-catenin-independent aspect of APC-Asef interactions in colorectal cancer, however, to the best of my knowledge however, it is not clear whether decreasing APC-Asef interactions in colorectal cancer cells will show no impact on the amounts/frequency of other protein-protein interactions influencing the canonical Wnt signaling pathway. The authors should at least test potential changes in nuclear/cytoplasmic b-catenin levels in response to MAIT-516, its level of ubiquitination/degradation, or changes in transcriptional activity (e.g. TOP flash luciferase assays). Moreover, the impact of their novel peptide on other critical functions, such as apoptosis, differentiation and tumorigenesis should also be assessed.

Response: Thank you for the suggestions. First, we explored whether MAIT-516 could influence canonical Wnt signaling. The protein half-life and the subcellular localization of β -catenin in SW480 cells treated with MAIT-516 were similar to those in mock-treated SW480 cells (Supplementary Fig. 14a-d). Consistently, quantitative polymerase chain reaction (qPCR) analysis revealed that the mRNA levels of β -catenin target genes (AXIN2, LGR5, CCND1, MYC) were not affected by MAIT-516 treatment in SW480 cells (Supplementary Fig. 14e-h). These data suggested that diminishing the APC-Asef PPI did not affect canonical Wnt signaling in CRC cells with

truncated APC expression. Second, we performed cell apoptosis assays with hIEC-6 normal intestinal cells and SW480 CRC cells treated with MAIT-516. In agreement with the CCK8 assay results, MAIT-516 did not induce the apoptosis of either hIEC6 or SW480 cells (Supplementary Fig. 16a, b). Finally, we evaluated the potential effect of MAIT-516 on intestinal cell differentiation. Notch inhibition by dibenzazepine (DBZ) allows colorectal stem cells to differentiate into goblet cells (*Nature*, **2005**, 435: 959-563). The DBZ-induced goblet cell differentiation in LS-174T cells is a well-established intestinal differentiation model (*Nat Cell Biol*, **2015**, 17: 7-19). We observed that NOTCH inhibition by DBZ significantly suppressed the mRNA level of the NOTCH target gene HES1, while increasing the mRNA levels of goblet cell markers (MUC2, TFF3, ATOH1, SPDEF), which indicated induced goblet cell differentiation (Supplementary Fig. 16c). Using the above system, we found that MAIT-516 did not affect the gene expression of goblet cell markers which indicates no impact of MAIT-516 on the differentiation of intestinal cells. We have added the results in the revised manuscript (page 13, lines 285-292 and 301-306; page 14, lines 307-312).

Comment II. I understand that the authors are considering HEK293T cells as a normal cell model (see Figure S7 title). In many contexts in cell biology and drug discovery, HEK293T should not be used as a normal cell line, and I consider critical for the authors to assess the impact of MAIT-516 in a bona fide normal colon cell model vs. cancer cell lines, for migration and proliferation/growth assays. This could be done using primary colonic epithelial cells or established models described in the literature. This would be critical to confirm the cancer-specific relevance of MAIT-516 in the development of future therapeutic approaches.

Response: According to the suggestions, we assessed the impact of MAIT-516 in hIEC6 cell line (CRL-3266, ATCC) which is a widely used normal intestinal cell model (*Exp Cell Res*, **1996**, 224: 354-364). Same as the results in HEK293T cells, the cell migration, proliferation, and apoptosis of hIEC6 cells were not affected by MAIT-516 treatment (Supplementary Fig. 10, 15a, 16b). Furthermore, we assessed the toxicity of MAIT-516 in the primary organoids derived from mouse intestines and paired human normal and colorectal cancer tissues. MAIT-516 showed no cytotoxicity in these primary organoid models (Supplementary Fig. 15b-f). The low cytotoxicity and cancer-specific effect of MAIT-516 suggest the potential for future drug development. We have added the results in the revised manuscript (page 12, lines 266-268; page 13, lines 296-303).

Minor issues:

Comment 12. The paper needs a thorough proofreading and English editing job.

Response: We have carefully edited the language of our manuscript with the help of the English language services of American Journal Expert.

Comment 13. It looks odd to me to have data description with figure callouts in the introduction.

Response: We have moved the data description to the results part in the revised manuscript (page 5, lines 84-96).

Comment 14. In line 292 (Figure S7 legend) “CCK8 assays on cell proliferation after the treatment of MAI-516 in colorectal cancer cells and normal cells”: It should say “MAIT-516” instead of MAI-516. There’s also a typo in “normal”.

Response: We corrected the typo in the revised Supplementary Information (page 21, line 453-455).

REVIEWERS' COMMENTS

Reviewer #3 (Remarks to the Author):

The authors adequately addressed all major concerns raised during this round of revisions.

However, I would like to point out that the authors mentioned they used hIEC-6 cells as a human normal intestinal model, but in the supplemental figures and text section they reference it as "IEC-6". The IEC-6 line is a rat small intestinal model that is different from hIEC-6. Please clarify/fix this minor issue.

Response to Reviewer #3's comments:

Comment: The authors adequately addressed all major concerns raised during this round of revisions. However, I would like to point out that the authors mentioned they used hIEC-6 cells as a human normal intestinal model, but in the supplemental figures and text section they reference it as "IEC-6". The IEC-6 line is a rat small intestinal model that is different from hIEC-6. Please clarify/fix this minor issue.

Response: Thank you for the recommendation of our manuscript. Indeed, we used HIEC-6 cells as a human normal intestinal cell model to assess the effects of MAIT-516. We have changed hIEC-6 to HIEC-6 in the revised manuscript (page 12, line 268; page 13, lines 294, 302, and 304; page 19, lines 445 and 452; page 22, lines 504 and 515) and corrected the errors in the revised supplementary information (page 23, Supplementary Fig. 10; page 29, Supplementary Fig. 16b).